# The composition and organization of *Drosophila* heterochromatin are heterogeneous and dynamic

Joel M Swenson[1†], Serafin U Colmenares[1†], Amy R Strom[1,2], Sylvain V Costes[1], Gary H Karpen[1,2]*

[1]Division of Biological Systems and Engineering, Lawrence Berkeley National Laboratory, Berkeley, United States; [2]Department of Molecular and Cell Biology, University of California, Berkeley, Berkeley, United States

**Abstract** Heterochromatin is enriched for specific epigenetic factors including Heterochromatin Protein 1a (HP1a), and is essential for many organismal functions. To elucidate heterochromatin organization and regulation, we purified *Drosophila melanogaster* HP1a interactors, and performed a genome-wide RNAi screen to identify genes that impact HP1a levels or localization. The majority of the over four hundred putative HP1a interactors and regulators identified were previously unknown. We found that 13 of 16 tested candidates (83%) are required for gene silencing, providing a substantial increase in the number of identified components that impact heterochromatin properties. Surprisingly, image analysis revealed that although some HP1a interactors and regulators are broadly distributed within the heterochromatin domain, most localize to discrete subdomains that display dynamic localization patterns during the cell cycle. We conclude that heterochromatin composition and architecture is more spatially complex and dynamic than previously suggested, and propose that a network of subdomains regulates diverse heterochromatin functions.

*For correspondence: karpen@ fruitfly.org

†These authors contributed equally to this work

**Competing interests:** The authors declare that no competing interests exist.

## Introduction

Eukaryotic genomes are composed of cytologically and functionally distinct chromatin domains called heterochromatin and euchromatin (*Heitz, 1928*). Although heterochromatin is primarily comprised of simple repetitive DNA sequences (*Peacock et al., 1978*) and transposons (*Carlson and Brutlag, 1978*), this domain is necessary for organismal functions, including pericentromeric sister chromatid cohesion (*Bernard et al., 2001*), achiasmate chromosome pairing and segregation in male and female meiosis (*Dernburg et al., 1996*; *Karpen et al., 1996*; *McKee and Karpen, 1990*), and genome integrity (*Peng and Karpen, 2009*). Heterochromatin is defined molecularly by H3K9me2/3 (deposited by the Su(var)3–9 histone methyltransferase [*Schotta et al., 2002*]) and its highly conserved and essential (*Eissenberg et al., 1992*; *Aucott et al., 2008*) binding partner Heterochromatin Protein 1 (*Grewal and Jia, 2007*) (HP1).

To understand how HP1 is able to regulate diverse cellular and organismal functions (*Grewal and Jia, 2007*), researchers have affinity purified HP1 in human tissue culture lines (*Rosnoblet et al., 2011*; *Lechner et al., 2005*), *S. pombe* (*Motamedi et al., 2008*) and *D. melanogaster* (*Ryu et al., 2014*; *Alekseyenko et al., 2014*) and identified >100 putative HP1 interacting proteins (HPips) by mass spectrometry. However, the overlap between identified HPips in these studies is minimal. Possible explanations include overexpression of the bait, or isolation of different populations of HP1. Indeed, biochemical (i.e. salt fractionation and size exclusion chromatography) and cytological (i.e. fluorescence correlation spectroscopy and fluorescence recovery after photobleaching) experiments

**eLife digest** If the DNA in a single human cell is stretched from end to end it is about two meters long, yet it all fits into a space that is just six thousandths of a millimeter across. This feat is possible because protein complexes package the cell's DNA into a form called chromatin to make it more compact. One type of chromatin – called "heterochromatin" – is needed to ensure that the DNA is positioned properly inside the cell's nucleus and segregated correctly when the cell divides. Heterochromatin contains many repeated DNA sequences that are repressed or 'silenced', as well as some active genes. Though heterochromatin accounts for about 25% of the human genome, little is known about the basic molecular processes that occur in this type of chromatin. This is in part because it is not clear which proteins are present in heterochromatin or how these proteins contribute to its structure and roles within the cell.

Swenson, Colmenares et al. have now combined two different approaches to search for proteins that are present in heterochromatin and genes that are needed to regulate heterochromatin's structure. These searches were conducted using fruit fly cells grown in the laboratory, and identified 118 candidate proteins and 374 candidate genes.

Next, Swenson, Colmenares et al. looked more closely at 89 of the proteins and confirmed that 30 did indeed localize to heterochromatin. Unexpectedly, more detailed imaging studies showed that these proteins were often localized to restricted regions within heterochromatin (referred to as subdomains). This closer look also revealed that many of the subdomains are dynamic, because the proteins change where they are localized as the cells grow and divide. Finally, many of the candidate proteins were shown to alter the ability of heterochromatin to silence genes.

These findings identify a host of new proteins and genes that bind and regulate heterochromatin. More importantly, Swenson, Colmenares et al. reveal that heterochromatin is structurally complex and contains many dynamic, smaller subdomains. The next critical challenges are to find the molecular mechanisms responsible for this unusual organization and to explore the roles of individual heterochromatin proteins or subdomains.

suggest the presence of distinct HP1 complexes (*Rosnoblet et al., 2011*; *Müller et al., 2009*; *Schmiedeberg et al., 2004*; *Huang et al., 1998*; *Kellum et al., 1995*). Regardless, studies in *Drosophila* have shown that the founding HP1 ortholog (HP1a) physically interacts with chromatin (*Bannister et al., 2001*; *Lachner et al., 2001*; *Lu, 2013*), replication components (*Pak et al., 1997*; *Murzina et al., 1999*; *Pindyurin et al., 2008*), chromatin modifying proteins (*Schotta et al., 2002*; *Delattre et al., 2000*; *Ito et al., 2012*), mRNA processing proteins (*Piacentini et al., 2009*), telomere protection proteins (*Shareef et al., 2001*; *Raffa et al., 2009*; *Cenci et al., 2003*) and components of small RNAi pathways (*Brower-Toland et al., 2007*; *Yin and Lin, 2007*). Despite extensive information about HPips, most have not been demonstrated to directly regulate heterochromatin organization or functions, and it is unclear how HPips are organized and regulated within the heterochromatin domain.

Historically, polytenized salivary gland chromosomes have been used to determine the localization of chromatin-bound proteins in *Drosophila*. However, the size of the heterochromatin domain in these terminally differentiated cells is reduced due to severe underreplication of heterochromatic repeats (*Rudkin, 1969*), which limits the resolution of HPip localization patterns within heterochromatin. Nevertheless, ATF-2 (*Seong et al., 2011*) and PIWI (*Brower-Toland et al., 2007*) were shown to occupy restricted regions or subdomains within the entire heterochromatin domain (hereafter 'holodomain') in polytene nuclei, suggesting that heterochromatin may be compartmentalized into functional units. However, the generality of subdomain organization for heterochromatin proteins is unknown, especially in mitotically dividing diploid cells.

One known function of heterochromatin domains is epigenetic transcriptional silencing of repeated DNAs (*Sienski et al., 2012*) and developmentally-regulated protein-coding genes (*Clowney et al., 2012*). Position effect variegation (PEV) describes the mosaic expression of euchromatic genes relocated or inserted in or near heterochromatin, which results from spreading of repressive heterochromatic components and clonal inheritance of the silenced state (reviewed in

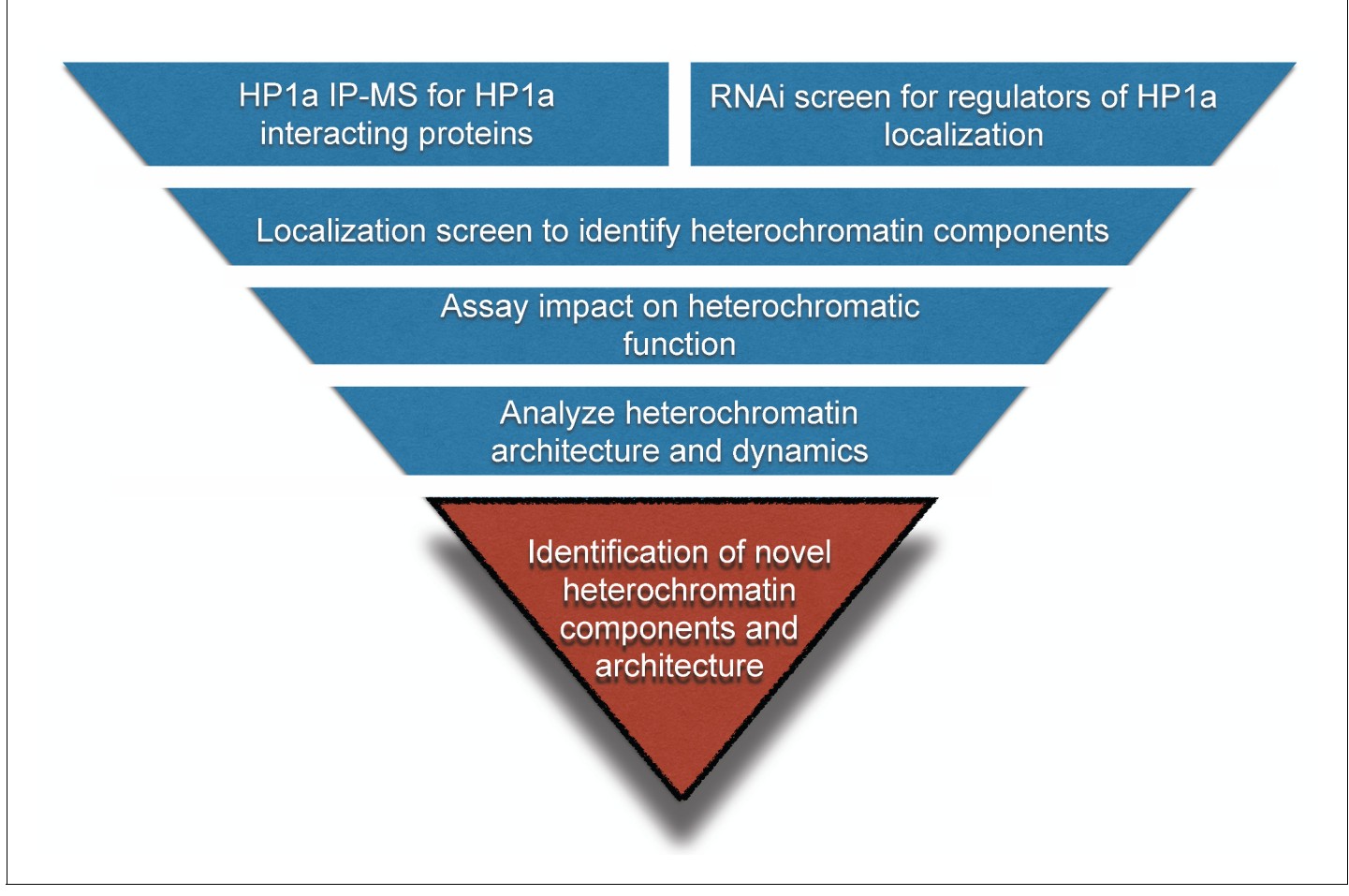

**Figure 1.** Workflow to identify novel heterochromatin components and regulators. We devised an unbiased strategy to identify novel components of heterochromatin. First, we identified candidates by performing HP1a immunoprecipitation followed by mass spectrometry (IP-MS) and a genome-wide RNAi screen. Candidates that localized to heterochromatin were assayed for effects on PEV. Finally, we investigated their spatial and temporal localization with respect to heterochromatin.

[**Wakimoto, 1998**]). Modification of PEV has been used as a sensitive assay to identify gene products that regulate heterochromatin structure and function (**Lewis, 1950**). For example, loss-of-function mutations in HP1a act as dominant suppressors of PEV (Su(var)) (**Eissenberg et al., 1990**; **Sinclair et al., 1983**), resulting in reduced repression, whereas increased HP1a levels result in enhancement of PEV (E(var), increased repression) (**Eissenberg et al., 1992**). Forward genetic screens in *Drosophila* have identified ~500 dominant mutations (estimated to map to 150 genes) that can modify PEV, however only ~30 have been mapped to specific genes thus far (**Elgin and Reuter, 2013**).

To gain further insight into the organization and function of heterochromatin (**Figure 1**), we implemented two approaches: 1) a biochemical purification of HP1a to identify novel binding partners, and 2) an image-based genome-wide RNAi screen to identify new regulators of HP1a levels and organization. Image analysis of a subset of candidates from both screens identified 30 proteins that localize to heterochromatin. The majority of these suppressed PEV when mutated or depleted by RNA interference (RNAi), demonstrating impact on heterochromatin-mediated gene silencing. Most importantly, more detailed imaging studies showed that both novel and previously known heterochromatin proteins are predominantly localized to restricted subdomains within heterochromatin, and display diverse, dynamic localization patterns during the cell cycle. In addition to greatly expanding our understanding of the number and types of heterochromatin proteins and regulators,

**Table 1.** HP1a interactors ranked by frequency of detection. The most common HP1a interacting proteins are listed according to the frequency in which they were detected in HP1a IP-MS experiments (out of six). References that link a protein to HP1a by IP, yeast-two-hybrid or immunofluorescence are listed in the third column. Asterisk indicates that the protein has been shown to modulate PEV. See **Table 1—source data 1** and **2** for a complete list of hits and **Table 1—source data 3** for a silver-stained gel of the IP.

| Flybase Gene Name | # of experiments enriched in | Literature Linking the Gene to HP1a |
|---|---|---|
| ADD1* | 6 | *Alekseyenko et al., 2014* |
| CG8108 | 6 | *Alekseyenko et al., 2014*; *Guruharsha et al., 2011* |
| HP5* | 6 | *Greil et al., 2007*; *Alekseyenko et al., 2014* |
| Su(var)3-9* | 6 | *Schotta et al., 2002*; *Alekseyenko et al., 2014* |
| Su(var)2-HP2* | 6 | *Shaffer et al., 2002*; *Alekseyenko et al., 2014* |
| tsr | 6 | |
| Hsc70-4 | 5 | *Alekseyenko et al., 2014* |
| Kdm4A* | 5 | *Lin et al., 2008*; *Alekseyenko et al., 2014*; Colmenares et al., unpublished |
| Odj (CG7357) | 5 | *van Bemmel et al., 2013* |
| smt3 | 4 | *Alekseyenko et al., 2014* |
| Lhr | 4 | *Greil et al., 2007*; *Alekseyenko et al., 2014* |
| Act5C | 4 | |
| Hsc70-3 | 4 | |
| betaTub56D | 4 | |
| Chd64 | 4 | |
| Hsp83 | 4 | |
| CG7692 | 3 | *Alekseyenko et al., 2014* |
| HP4* | 3 | *Greil et al., 2007* |
| Tudor-SN | 3 | |
| His2B:CG33872 | 3 | |
| eIF-4a | 3 | |
| FK506-bp1 | 3 | |
| CG7172 | 3 | |
| CG8258 | 3 | |
| EF2 | 3 | |
| eIF-4B | 3 | |
| Hsc70-5 | 3 | |
| Hsp60 | 3 | |
| qm | 3 | |
| sta | 3 | |

Source data 1. 2-Step HP1a IP-MS. HP1a was purified in the absence of ionizing radiation (IR) (A), and 10 min (B) and 60 (C) minutes after 10 Gy exposure. Number of unique peptides per protein are listed. The HPips identified did not change significantly with respect to irradiation, therefore we used all purifications to identify candidate hits.

Source data 2. 1-Step HP1a IP-MS. HP1a was purified in the absence of IR (A, Mock and FS-HP1a), and 10 (B, FS-HP1a) and 60 (C, FS-HP1a) minutes after 10 Gy exposure. Number of unique peptides per protein is listed. The HPips identified did not change significantly with respect to irradiation, therefore we used all purifications to identify candidate hits.

Source data 3. HP1a interacts with a large set of proteins. Silver-stained gel of a single step purification from S2 cells stably expressing FS-HP1a (lanes 1–3) or WT (lane 4) S2 cells. HP1a was purified in the absence of IR (lane 1), and 10 (lane 2) and 60 (lane 3) minutes after 10 Gy exposure. The HPips identified did not change significantly with respect to irradiation, therefore all purifications were used to identify candidate hits.

these findings lead us to propose that heterochromatin is composed of a dynamic network of subdomains that regulates different heterochromatin functions.

## Results

### IP-MS identification of HP1a interaction partners reveals new candidate heterochromatin components

To gain a better understanding of the composition of heterochromatin we purified HP1 six independent times, from S2 cells stably expressing HP1a tagged with 3X-FLAG and StrepII (FS-HP1a) at ~20% of endogenous HP1a levels (data not shown). Purified samples were analyzed by liquid chromatography-tandem mass spectrometry (LC-MS/MS). MS results identified 135 proteins that were significantly enriched in at least two of the six IP-MS experiments (*Table 1* and *Table 1—source data 1* and *2*) (hereafter HPips). To investigate the potential biological functions of these proteins we used the Database for Annotation, Visualization and Integrated Discovery (DAVID) v6.7 (*Huang et al., 2008*, *2009*) toolset to identify enriched gene ontology (GO) terms. Consistent with expectations for heterochromatic proteins, these HPips were enriched for GO categories that include 'chromocenter', 'chromatin organization', 'chromatin assembly or disassembly' and 'post-transcriptional regulation of gene expression' (*Supplementary file 1*). Initial validation of the approach comes from the observation that the 135 candidate HP1a interactors included 17 of the ~33 previously characterized HPips (~52%), such as HP2 (*Shaffer et al., 2002*), Lhr (*Greil et al., 2007*), HP4 (*Greil et al., 2007*), HP5 (*Greil et al., 2007*), Su(var)3–9 (*Schotta et al., 2002*) and Kdm4a (*Lin et al., 2008*) (*Table 1—source data 1* and *2*).

Most importantly, 118 of the HPips isolated here (89%) were not previously identified as *Drosophila* HP1a interactors. Five of these new HPips were previously identified as Su(var)s, demonstrating their functional importance to heterochromatin (Nap1 [*Stephens et al., 2006*], Hel25E [*Eberl et al., 1997*], His2Av [*Swaminathan et al., 2005*], Pp1-87B [*Reuter et al., 1990*], and RpLP0 [*Frolov and Birchler, 1998*]; *Table 1* and *Table 1—source data 1* and *2*). The remaining 113 HPips were not previously shown to impact heterochromatin functions or associate with HP1a, and potentially represent a large collection of novel heterochromatin components.

### Image-based genome-wide RNAi screen identifies new candidate regulators of HP1a recruitment or maintenance

In order to identify factors that regulate heterochromatin independent of HP1a binding, we performed an image-based genome-wide RNAi screen (*Figure 2A*) for gene depletions that altered heterochromatin architecture (e.g. HP1a levels or localization). Nuclei were identified based on DAPI staining, and analyzed for 33 different imaging features (e.g. nuclear size, nuclear shape, channel-specific intensity/distribution metrics: see *Supplementary file 2*). To address known issues associated with genome-wide screens (e.g. biological noise, transfection efficiency, image quality) we employed positive (HP1a dsRNA) and negative (GFP dsRNA) controls, performed the screen in duplicate and utilized Rank Product normalization (*Breitling et al., 2004*), which incorporates replicate consistency and provides an estimated p-value for observed differences. We utilized three different candidate identification methods (rank lists of individual features of interest, supervised and unsupervised clustering, see 'Materials and methods') to maximize the number of true positive hits.

First, we focused on the identification of genes whose absence results in reduced HP1a fluorescence, or phenocopies HP1a depletion (hereafter HP1a positive regulators, HPprs). Of the 374 genes identified as putative hits (*Figure 2B* and *Figure 2—source data 1*), 22 were previously implicated in regulating HP1a localization or heterochromatin properties (e.g. PEV) (*Table 2*). Notably, of the 374 HPprs, only three genes besides HP1a (Tudor-SN, RpL8 and mRpL3) were also identified in the HP1a IP-MS. This suggests that the majority of HPprs are not tightly bound to HP1a, as they do not co-isolate, and may indirectly influence HP1a establishment or maintenance. Second, we identified 564 genes (including 8 that were identified in the HP1a IP-MS) that negatively regulate HP1a fluorescence intensity (i.e. HP1a fluorescence is increased in their absence) (*Supplementary file 3*). We speculate that some of these 564 genes may normally be required for removal/turnover of HP1a, but are not investigated further here.

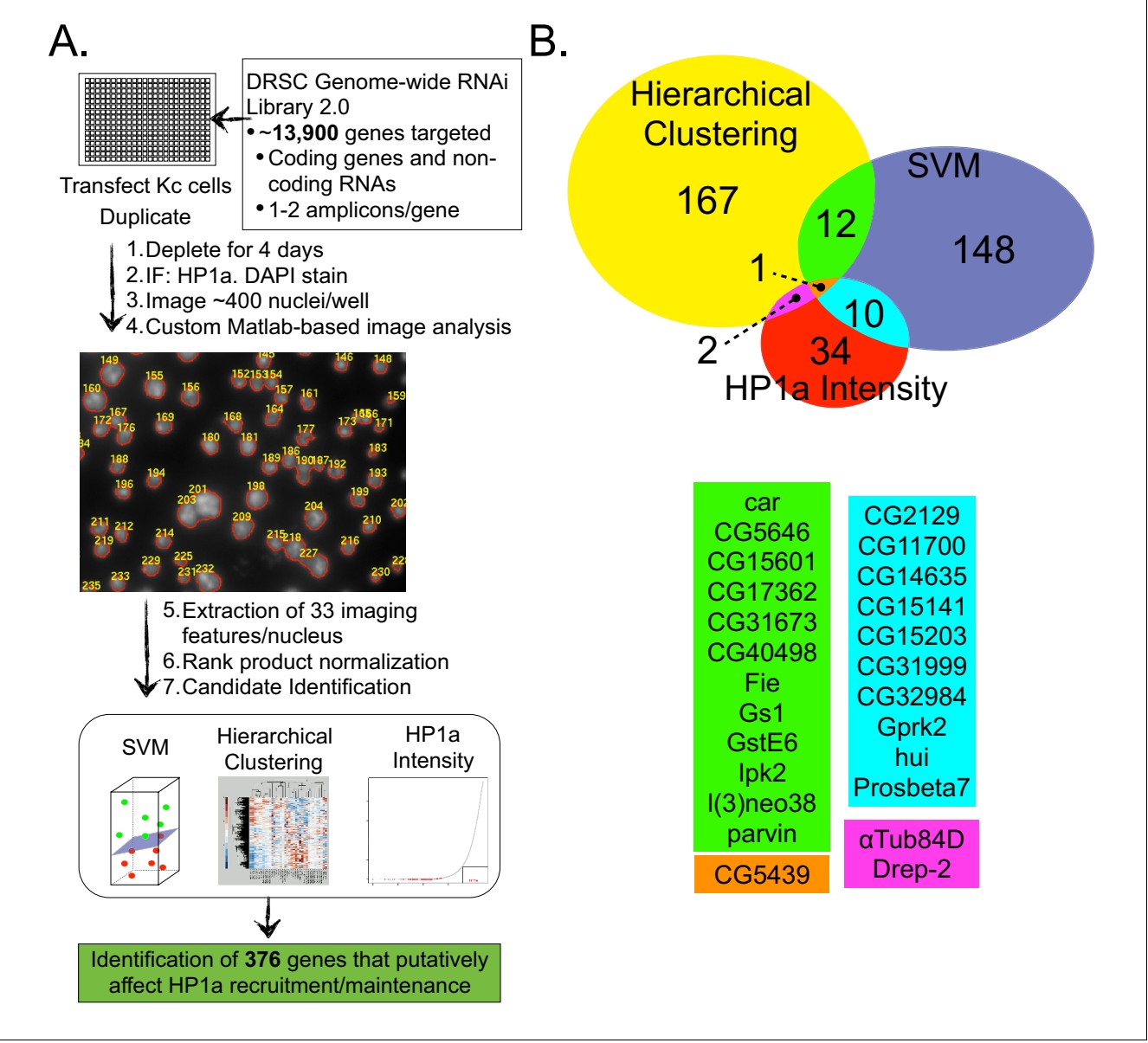

**Figure 2.** A genome-wide image-based RNAi screen identifies HP1a regulators. Drosophila Kc cells transfected with dsRNA were analyzed for HP1a localization by IF, and DNA was counterstained with DAPI. Cells were visualized using high-throughput fluorescent microscopy and imaging features were extracted using custom Matlab scripts. Wells were normalized and checked for replicate consistency using the Rank Product test and a p-value was calculated. Putative candidates involved in HP1a recruitment/maintenance were selected by identifying amplicons that lowered HP1a intensity, or clustered with HP1a depletions after hierarchical clustering or Support Vector Machine (SVM) analysis. (B) Genes that clustered using unsupervised hierarchical clustering with either HP1a or Su(var)3–9 positive control depletions are represented by the yellow circle. Supervised machine learning models (SVMs) were trained to identify genes that disrupt HP1a staining (blue circle) using HP1a depletion controls. HP1a intensity measures (mean, maximum, relative maximum and kurtosis) were used to identify another set of candidate genes (red circle). Genes identified by multiple methods are indicated by color below the Venn diagram. See *Figure 2—source data 1* for a list of all genes identified in the RNAi screen and the method used to identify them.

The following source data and figure supplement are available for figure 2:

**Source data 1.** 374 genes putatively regulate heterochromatin.

**Figure supplement 1.** The rank product test is more effective than the robust Z-Score at identifying HP1a knockdowns.

Consistent with our expectations, GO terms analysis of all HPrs identified enrichment for genes associated with the chromocenter, chromatin, RNA interference, RNA-binding and sequence-specific DNA binding (*Supplementary file 4*). The identification of genes associated with GTP binding, the proteasome, response to heat and glutathione metabolism is unexpected and may represent noise. However, the correct identification of 22 (*Table 2*) known regulators and the high accuracy in

**Table 2.** RNAi screen hits with previously known connections to heterochromatin. Identified hits from the RNAi screen with previously known connections to heterochromatin are listed according to the method of identification (Hierarchical Clustering, HP1a Intensity or Support Vector Machine [SVM]). Whether a gene clustered with HP1a or Su(var)3–9 depletion controls after Hierarchical Clustering is indicated in parentheses.

| Flybase Name or Symbol | Method of Identification | Link to Heterochromatin | Reference |
|---|---|---|---|
| Ssrp | Hierarchical Clustering (HP1a, Su(var)3-9) | Part of FACT complex | *Orphanides et al., 1999* |
| MBD-like | Hierarchical Clustering (HP1a) | Repressive, localizes to chromocenter, part of NuRD complex | *Ballestar et al., 2001*; *Marhold et al., 2004* |
| stellate | Hierarchical Clustering (HP1a) | Subunit of Casein kinase II | *Bozzetti et al., 1995* |
| kismet | Hierarchical Clustering (HP1a) | Su(var), regulates heterochromatic silencing | *Schneiderman et al., 2009*, *2010* |
| Spt20 | Hierarchical Clustering (HP1a) | Part of SAGA complex | *Weake et al., 2009* |
| Su(var)205 | Hierarchical Clustering (HP1a), HP1a Intensity, SVM | Encodes HP1a | |
| l(3)neo38 | Hierarchical Clustering (HP1a), SVM | Regulates heterochromatic silencing | *Schneiderman et al., 2010* |
| Hdac3 | Hierarchical Clustering (Su(var)3-9) | Ortholog regulates HP1beta levels | *Bhaskara et al., 2010* |
| Rm62 (lip, p68) | Hierarchical Clustering (Su(var)3-9) | Su(var), binds and putatively targets Su(var)3-9, binds blanks, binds AGO2, regulates heterochromatic silencing | *Csink et al., 1994*; *Boeke et al. 2011*; *Gerbasi et al., 2011*; *Ishizuka et al., 2002*; *Schneiderman et al., 2010* |
| jumu | Hierarchical Clustering (Su(var)3-9) | Localizes to chromocenter, modifier of variegation | *Hofmann et al., 2010*, *2009* |
| MTA1-like | Hierarchical Clustering (Su(var)3-9) | Part of NuRD complex | *Marhold et al., 2004* |
| AGO2 | Hierarchical Clustering (Su(var)3-9) | Heterochromatin targeting, Su(var) | *Noma et al., 2004*; *Deshpande et al., 2005* |
| moi | Hierarchical Clustering (Su(var)3-9) | Protects telomeres | *Raffa et al., 2009* |
| Adar | HP1a Intensity | E(var) on the 4th chromosome, edits RNA, silences *Hoppel's* transposase | *Savva et al., 2013* |
| Parp | HP1a Intensity | E(var), promotes chromatin condensation and represses retrotransposons | *Tulin and Spradling, 2003* |
| Ino80 | HP1a Intensity | Ortholog in mice complexed with YY1 which regulates HP1gamma, regulates heterochromatic silencing | *Wu et al., 2009*; *Schneiderman et al., 2010* |
| roX1 | HP1a Intensity | Su(var) | *Deng et al., 2009* |
| modulo | SVM | Localizes to chromocenter, Su(var) | *Perrin et al., 1998*; *Garzino et al., 1992* |
| blanks | SVM | Regulates heterochromatic silencing | *Schneiderman et al., 2010* |
| crol | SVM | Regulates heterochromatic silencing | *Schneiderman et al., 2010* |
| Samuel | SVM | Regulates heterochromatic silencing | *Schneiderman et al., 2010* |
| Wapl | SVM | Su(var) | *Verni et al., 2000* |

identifying positive controls (*Figure 2—figure supplement 1*) suggests these categories may represent novel modes of regulating HP1a protein levels and/or distribution in the nucleus.

## A subset of IP-MS and RNAi hits colocalize with HP1a

We validated the heterochromatin association of HPips and HPprs by determining if the proteins colocalize with HP1a in S2 tissue culture cells. IP-MS candidates were selected for imaging if they had at least two unique peptides and a 3-fold enrichment over control. Common contaminants were eliminated (e.g. ribosomal and tubulin proteins [*Mellacheruvu et al., 2013*]) as were proteins previously known to colocalize with HP1a (e.g. KDM4A [*Lin et al., 2008*], HP4 [*Greil et al., 2007*]). RNAi screen candidates were chosen based on identification by more than one method (HP1a intensity metrics, supervised clustering [Support Vector Machine or SVM], or unsupervised clustering [hierarchical], or GO terms enrichment [sequence-specific DNA binding, RNA-binding, RNA interference, response to heat, chromatin organization]). The candidate list was further refined based on the availability of clones from the Berkeley *Drosophila* Genome Project (*Yu et al., 2011*). For genes with multiple isoforms, the gene isoform predominantly expressed in S2 cells according to published stranded RNA-seq data (*Brown et al., 2014*) was chosen.

Based on these criteria, we subcloned 89 unique protein-coding open reading frames (ORFs) (44 identified by HP1a IP-MS only, 44 by RNAi screen only, 1 from both HP1a IP-MS and RNAi screen) into a GFP expression vector and analyzed colocalization with mCherry-HP1a (*Figure 3*) by calculating the Pearson correlation coefficient (*Costes et al., 2004*) (PCC). Low-resolution/high-throughput imaging identified 30 candidates (34% of the 89) that colocalized with HP1a (see 'Materials and

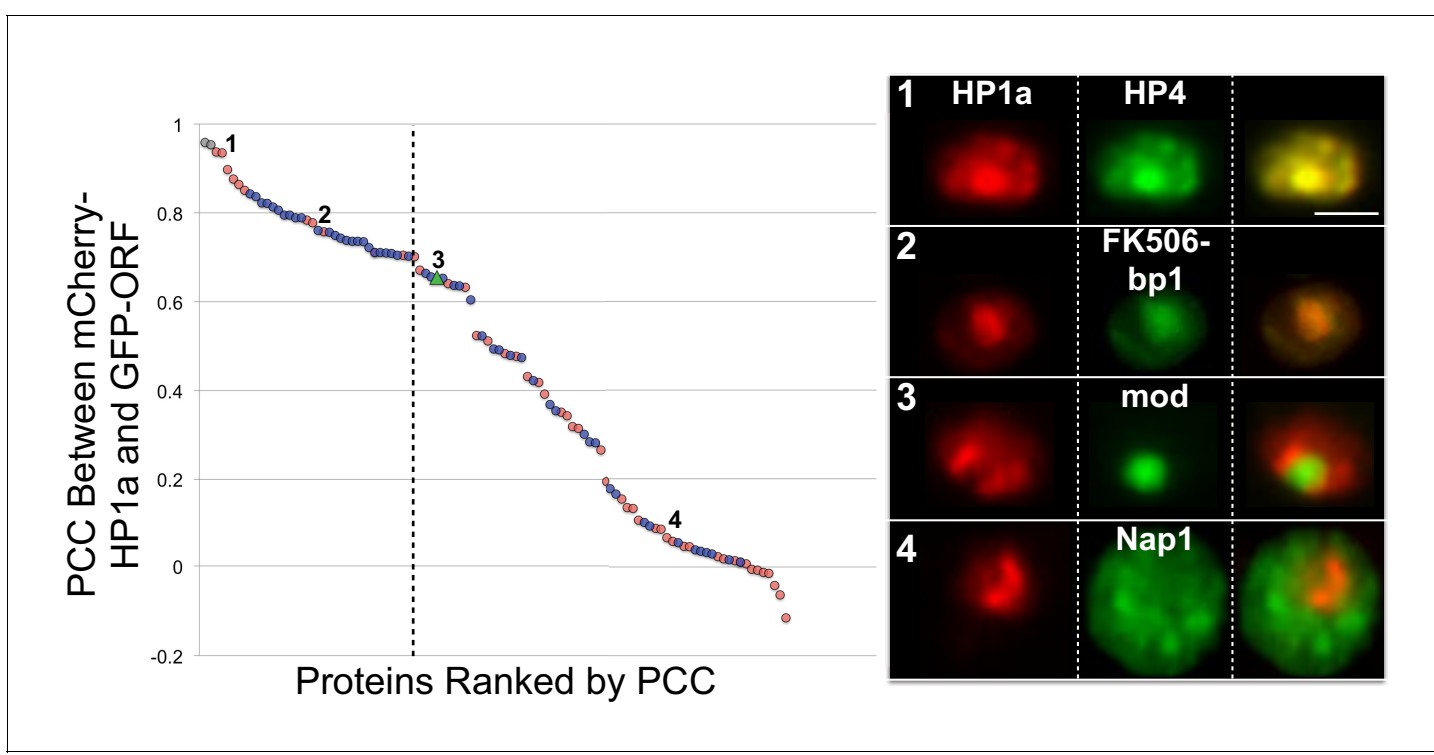

**Figure 3.** Identification of candidates that co-localize with HP1a. Proteins were selected from the HP1a IP-MS (red circles) or the RNAi screen (blue circles), tagged with GFP (green), and analyzed for localization with respect to mCherry-tagged HP1a (red). GFP-tagged HP1a was used as a positive control (gray circles). The Pearson correlation coefficient (PCC) between mCherry-HP1a and GFP-tagged proteins left of the dashed line was significantly higher than the PCC between mCherry-HP1a and GFP-mod (green triangle), using the two-sided unpaired Mann-Whitney test (p-value<0.05). Numbers on graph correspond to representative images (right panel). Scale bar is 5 µm. See *Figure 3—source data 1* for the PCC of all proteins tested.

The following source data is available for figure 3:

**Source data 1.** Identification of candidates that co-localize with HP1a.

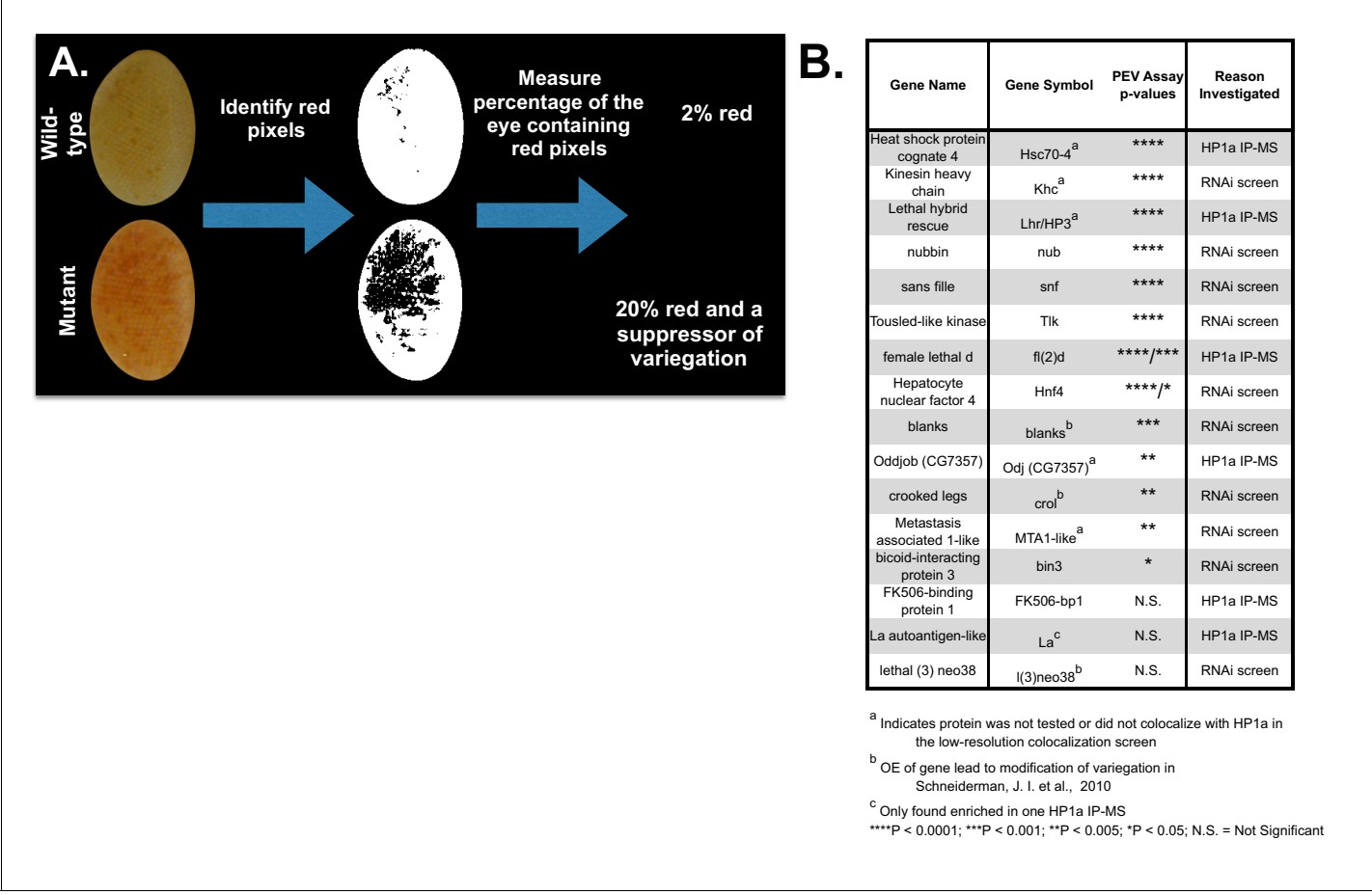

**Figure 4.** HPips and RNAi screen candidates are suppressors of variegation. (**A**) Color Inspector 3D in ImageJ was used to determine the RGB values of 'red' pixels (indicating loss of suppression). The percent of the eye composed of red pixels was then calculated. (**B**) Fly mutants and RNAi lines were tested for impact on white variegation in *y, w, KV108* males, and are organized by p-value. Mutations were tested for dominant effects if they were recessive lethal, otherwise homozygotes were analyzed. CG7357[f00521] was scored for variegation using the yellow reporter gene, since the line harbors a mini-white reporter that precludes assessment of white variegation. The p-values were calculated using a 2-tailed, 2-sample unequal variance t-test for white variegation and a 2-sample Kolmogorov-Smirnov test for yellow variegation. Positive and negative controls were performed and are listed in the *Figure 4—source data 1* along with the genotypes of all the fly lines used. CG2129, Ssrp and Ref1 could not be tested for effects on variegation using RNAi lines, due to lethality.

The following source data is available for figure 4:

**Source data 1.** HPips and RNAi screen candidates are suppressors of variegation.

methods'), 9 of which were identified by HP1a IP-MS (9/44=20%) and 21 from the RNAi screen (21/44=48%) (*Figure 3* and *Figure 3—source data 1*). We conclude that ~1/3 of the tested candidates are likely to be physically associated with the heterochromatin domain, and are analyzed in more detail below. The remainder were not localized to heterochromatin due to technical reasons (e.g. poor expression or produced non-functional proteins), or could regulate HP1a/heterochromatin indirectly or represent noise from the screens, and were not studied further.

## Many HPips and HPprs affect heterochromatin-mediated silencing

Transcriptional silencing is a defining feature of heterochromatin and screens for dominant modifiers of PEV have identified many key heterochromatin components. To determine if proteins that colocalized significantly with HP1a (*Figure 3* and *Figure 3—source data 1*) affect heterochromatin properties in the fly, we assayed publicly available fly mutants or RNAi lines for modification of PEV (using

either *yellow⁺* or *white⁺* reporter genes) (see 'Materials and methods' for details). We tested 11 different HPips or HPprs for mutant or depletion effects on PEV, and found that 8 suppress PEV (*Figure 4* and *Figure 4—source data 1*). The high success rate at identifying modifiers of PEV (8/11 tested = 73%) suggests that most candidates that colocalize with HP1a (*Figure 3* and *Figure 3— source data 1*) are also likely to impact heterochromatin properties. We extended the PEV analysis to 5 other candidates identified as HPips (CG7357, Lhr/HP3) or HPprs (MTA1-like, Khc and Hsc70-4) (*Table 1* and *Alekseyenko et al., 2014*; *Greil et al., 2007*; *van Bemmel et al., 2013*), whose colocalization with HP1a was not determined (above). Mutant alleles/RNAi lines for all 5 candidates produced a Su(var) phenotype (*Figure 4* and *Figure 4—source data 1*). Altogether, fly mutants or RNAi lines targeting 13 of 16 tested candidates (81%) produced a Su(var) phenotype (*Figure 4* and *Figure 4—source data 1*). We conclude that the multi-pronged experimental approach (*Figure 1*) was very efficient at identifying functional heterochromatin components. In addition, since ~30 modifiers of variegation were previously mapped to specific genes (*Elgin and Reuter, 2013*), this represents an ~50% increase in the number of known proteins that regulate PEV. Given that CG7357 localizes to heterochromatin and is required for silencing, we propose naming the gene 'Oddjob' (Odj).

## Localization of IP-MS and RNAi hits reveals complex patterns within heterochromatin

Indiscriminate binding of HPips or HPprs to HP1a predicts that these proteins should be broadly distributed across the entire heterochromatin holodomain. However, HPips or HPprs could selectively bind to different HP1a populations, or directly bind specific repeated sequences, resulting in localization to restricted compartments or subdomains of the heterochromatin holodomain. Additionally, we suspected that implementation of the PCC to define colocalization with HP1a may have led to the identification of false-positives. Therefore, to validate and more precisely determine candidate protein localization patterns in heterochromatin, we analyzed a subset (19) of the top HP1a colocalization hits (30) using higher-resolution microscopy and manual curation of the higher-resolution localization patterns (see 'Materials and methods'). We successfully validated heterochromatin localization for 7 of the 9 strongest colocalizers (PCC > 0.79), and 12/19 total (*Table 3* and *Figure 3— source data 1*). Surprisingly, we found that most HPips displayed restricted patterns within heterochromatin. Four general patterns were observed (*Figure 5* and *Table 3*): 1) broad – near-complete overlap with HP1a (e.g. HP4); 2) narrow – significantly less overlap with HP1a, especially at the periphery of HP1a domains (e.g. FK506-bp1); 3) focal – one or a few highly restricted foci embedded in or adjacent to HP1a (e.g. crol); and 4) at the heterochromatin boundary – partial overlap with the edge of the HP1a domain (e.g. Hrb87F, Tlk; *Figure 5—figure supplement 1*). Hereafter we refer to the narrow, focal and boundary classes as subdomain-forming proteins.

## Identification of complex heterochromatin patterns for known heterochromatin components

To better understand the spatial organization of the heterochromatin domain and evaluate the generality of subdomain architecture, we localized 12 previously identified HPips and repeated-DNA binding proteins at high-resolution in S2 cells (see *Table 3* for summary; see *Figure 5* and *Figure 5— figure supplement 1* for images). Broad colocalization with HP1a was verified for HP5 (*Greil et al., 2007*), Kdm4a (Colmenares et al., unpublished), Su(var)3–7 (*Cléard et al., 1997*), and Su(var)3–9 (*Schotta et al., 2002*). However, other proteins displayed more complex patterns within the heterochromatin of S2 cells than were previously reported using cells containing polytenized chromosomes (*Alekseyenko et al., 2014*; *Pindyurin et al., 2008*; *Shaffer et al., 2002*, *2006*). For example, SuUR occupies a narrow subdomain within the holodomain, and HP2 forms a narrow subdomain enriched at the domain boundary. Interestingly, ADD1 isoform A forms a narrow subdomain within the holodomain, while ADD1 isoform B (ADD1-PB) occupies a focal subdomain at the domain boundary. Overall, 7 of the 12 previously identified HPips and repeated-DNA binding proteins were classified as forming subdomains (*Table 3*).

We conclude that protein localization within heterochromatin is more complex and diverse than previously recognized. We observed proteins that exhibited broad (10 proteins), narrow (7), focal (11) and heterochromatin boundary (8) patterns (*Table 3*), which were not mutually exclusive (see below). The identification of a large number of subdomain-forming HPips (17/22 heterochromatin

**Table 3.** Localization patterns of known heterochromatin components, IP-MS and RNAi screen hits. Top candidates from the localization screen and proteins with a previously known connection to HP1a were imaged at higher resolution and grouped into four categories of heterochromatin localization, based on live imaging in the presence of fluorescently tagged HP1a: broad, narrow, focal, or at the heterochromatin boundary. Localization outside of heterochromatin is also noted. Proteins are sorted by their observed localization patterns. HC = heterochromatin, NR = nucleolar, EC = euchromatin, CP = cytoplasmic.

| Gene Name | Isoform | Reason Investigated | Heterochromatic Localization | | | | Other Localization Notes | | Previous Published Localization | Effect on Variegation |
|---|---|---|---|---|---|---|---|---|---|---|
| | | | Broad | Narrow | Focal | At HC Boundary | Pan Nuclear | Other | | |
| Heterochromatin protein 4 | HP4-RA | HP1a IP-MS | X | | | | | | Kc chromocenter (*Greil et al., 2007*) | Su(var) (*Greil et al., 2007*) |
| Heterochromatin protein 5 | HP5-RA[¶] | HP1a IP-MS | X | | | | | | Kc chromocenter (*Greil et al., 2007*) | Su(var) (*Greil et al., 2007*) |
| Lysine (K)-specific demethylase 4A | Kdm4A-RA[¶] | HP1a IP-MS | X | | | | | | Kc, S2 and BG3 chromocenter (Colmenares et al., unpublished) | Su(var) (Colmenares et al., unpublished) |
| Suppressor of variegation 3-9 | Su(var)3-9-RA[‡,¶] | HP1a IP-MS | X | | | | | | polytene chromocenter (*Schotta et al., 2002*) | Su(var) (*Reuter et al., 1986*) |
| Suppressor of variegation 3-7 | Su(var)3-7-RB[†,¶] | literature | X | | | | | | polytene chromocenter, HC in embryos (*Cleard et al., 1997*) | Su(var) (*Reuter et al., 1990*) |
| Lethal hybrid rescue | Lhr-RA/HP3-RA[¶] | HP1a IP-MS | X | | X | | | | centromeric (*Thomae et al., 2013*); polytene chromocenter (*Brideau et al., 2006*); Kc chromocenter (*Greil et al., 2007*) | Su(var) (this study) |
| Heterochromatin protein 6 | HP6-RA[¶] | literature | | | X | | | Slight narrow HC enrichment | Kc chromocenter (*Greil et al., 2007*); polytene chromocenter (*Joppich et al., 2009*); Kc cells - centromeric (*Ross et al., 2013*) | Not a mod(var) (*Greil et al., 2007*); deficiency spanning gene is a Su(var) (*Doheny et al., 2008*) |
| Oddjob (CG7357) | Odj-RA[¶] | HP1a IP-MS | X | X | X | | | | - | Su(var) (this study) |
| Su(var)2-HP2 | Su(var)2-HP2-RB | HP1a IP-MS | X | X | | X | | | polytene chromocenter (*Shaffer et al., 2002*) | Su(var) (*Shaffer et al., 2002*) |
| blanks | blanks-RA* | RNAi screen | X | | X | | X | Foci outside HC | pan-nuclear (structured) (*Gerbasi et al., 2011*) | Su(var) (this study); OE mod(var) (*Schneiderman et al., 2010*) |
| CG2129 | CG2129-RA* | RNAi screen | X | | | X | | Foci outside HC | - | RNAi lines were lethal |
| FK506-binding protein 1 | FK506-bp1-RA | HP1a IP-MS | | X | | | | Foci outside HC | nucleolar based on DAPI-staining (*Edlich-Muth et al., 2015*) | Non-mod(var) (this study) |
| XNP | XNP-RA[¶] | literature | | X | X | | | | active genes and satellite DNA near HC in polytenes and imaginal discs (*Schneiderman et al., 2009*); Broad HC in polytenes (*Bassett et al., 2008*); Beta-heterochromatin of the X chromosome in polytenes (*Emelyanov et al., 2010*) | OE mod(var) (*Schneiderman et al., 2009*); Su(var) (*Bassett et al., 2008*), (*Emelyanov et al., 2010*) |
| Suppressor of Under-Replication | SuUR-RA[¶] | literature | | X | X | | | | polytene chromocenter (*Makunin et al., 2002*) | mutation is Su(var), extra copy is E(var): (*Belyaeva et al., 2003*) |
| Hormone receptor 83 | Hr83-RA*,[§] | RNAi screen | | X | | | X | NR | - | - |

*Table 3 continued on next page*

*Table 3 continued*

| Gene Name | Isoform | Reason Investigated | Heterochromatic Localization | | | At HC Boundary | Other Localization Notes | | Previous Published Localization | Effect on Variegation |
|---|---|---|---|---|---|---|---|---|---|---|
| | | | Broad | Narrow | Focal | | Pan Nuclear | Other | | |
| D1 chromosomal protein | D1-RA¶ | literature | | | X | | | Slight narrow HC enrichment | HC (SATI and SATIII) in embryos (*Aulner et al., 2002*) | Su(var) (*Aulner et al., 2002*) |
| lethal (3) neo38 | l(3)neo38-RB | RNAi screen | | | X | FOCI | | | - | Non-mod(var) (this study); OE mod(var) (*Schneiderman et al., 2010*) |
| crooked legs | crol-RD | RNAi screen | | | X | FOCI | | | nuclear (*Mitchell et al., 2008*) | Su(var) (this study); OE mod(var) (*Schneiderman et al., 2010*) |
| ADD domain-containing protein 1 | ADD1-RB | HP1a IP-MS | | | X | X | | Weak broad HC enrichment | polytene chromocenter (*Alekseyenko et al., 2014*) | Su(var) (*Alekseyenko et al., 2014*) |
| proliferation disrupter | prod-RA¶ | literature | | | X | X | | | AATAACATAG in 3rd instar larvae brains (*Platero et al., 1998*) | - |
| Heterogeneous nuclear ribonucleoprotein at 87F | Hrb87F-RA§ | RNAi screen | | | | X | | | polytene chromocenter (*Piacentini et al., 2009*) | Su(var) (*Piacentini et al., 2009*) |
| Tousled-like kinase | Tlk-RF | RNAi screen | | | | X | | 1-2 foci per nuc. Often 1 focus is abutting HP1a | nuclear, but not chromatin bound (*Carrera et al., 2003*) | Su(var) (this study) |
| RNA and export factor binding protein 1 | Ref1-RA# | HP1a IP-MS | | | | | X | Slight HC enrichment | nuclear membrane and nucleoplasm (*Buszczak and Spradling, 2006*) | - |
| sans fille | snf-RA | RNAi screen | | | | | Except nucleolus | | nuclear (*Flickinger and Salz, 1994*) | Su(var) (this study) |
| Hepatocyte nuclear factor 4 | Hnf4-RA | RNAi screen | | | | | Except nucleolus | | nuclear (*Palanker et al., 2009*; *Gutzwiller et al., 2010*) | Su(var) (this study) |
| bicoid-interacting protein 3 | bin3-RA | RNAi screen | | | | | X | | - | Su(var) (this study) |
| Cullin 4 | Cul4-RA¶ | literature | | | | | X | | - | - |
| female lethal d | fl(2)d-RA | HP1a IP-MS | | | | | X | | non-uniform in nucleus (*Penn et al., 2008*) | Su(var) (this study) |
| jumeau | jumu-RA§ | RNAi screen | | | | | X | | polytene chromocenter (*Strödicke et al., 2000*) | Su(var) (*Strödicke et al., 2000*) |
| La autoantigen-like | La-RA# | HP1a IP-MS | | | | | | EC | nuclear (*Yoo and Wolin, 1994*) | Non-mod(var) (this study) |
| Structure specific recognition protein | Ssrp-RA | RNAi screen | | | | | | NR | nucleolar (*Hsu, et al., 1993*) | - |

*Protein localization is dependent on which terminus of the gene is GFP-tagged and/or cell-type

†Stable tagged Kc cell line

‡Transient transfection of BG3 cells

§Less than 1% of cells expressed the construct

#Proteins were only found enriched in one HP1a IP-MS

¶Proteins were not tested for colocalization with HP1a in the low-resolution colocalization screen

proteins tested, 77%) shows that binding to HP1a is predominantly restricted within the heterochromatin, and not indiscriminate. We hypothesize that an unknown mechanism restricts HPip localization within the HP1a/heterochromatin holodomain (see 'Discussion').

### Live imaging reveals that subdomain protein localization patterns are dynamic

HP1a displays dynamic behavior during the cell cycle, which is essential for error-free mitosis (*Hirota et al., 2005*; *Mateescu et al., 2004*) and replication of heterochromatin (*Quivy et al., 2008*). HP1a is largely removed from chromatin during mitotic prophase, reloads starting at anaphase/telophase and remains a discrete domain associated with chromosomes throughout interphase (*Kellum et al., 1995*). Additionally, the localization of proteins that bind specific satellite repeats (Prod and GAGA factor) is cell cycle regulated (*Platero et al., 1998*). Therefore, we used time-lapse microscopy to analyze cell cycle changes in the localization of 7 fluorescently-tagged HP1a interactors/regulators that exhibited multiple patterns in the previous analyses (*Table 3* and *Figure 5—figure supplement 1*), relative to the heterochromatin domain (HP1a-GFP).

The localization patterns were surprisingly dynamic, and in some cases suggest potential biological functions. For example, HP2 and SuUR both colocalize with PCNA foci (replication [*Moldovan et al., 2007*]) during early, mid and late S-phase (HP2, *Figure 6B* and *Video 1*; SuUR, *Figure 7—figure supplement 1*, *Video 2* and *Nordman et al., 2014*), suggesting links to replication. Indeed, SuUR prevents polytenization of heterochromatic sequences (*Belyaeva et al., 1998*)

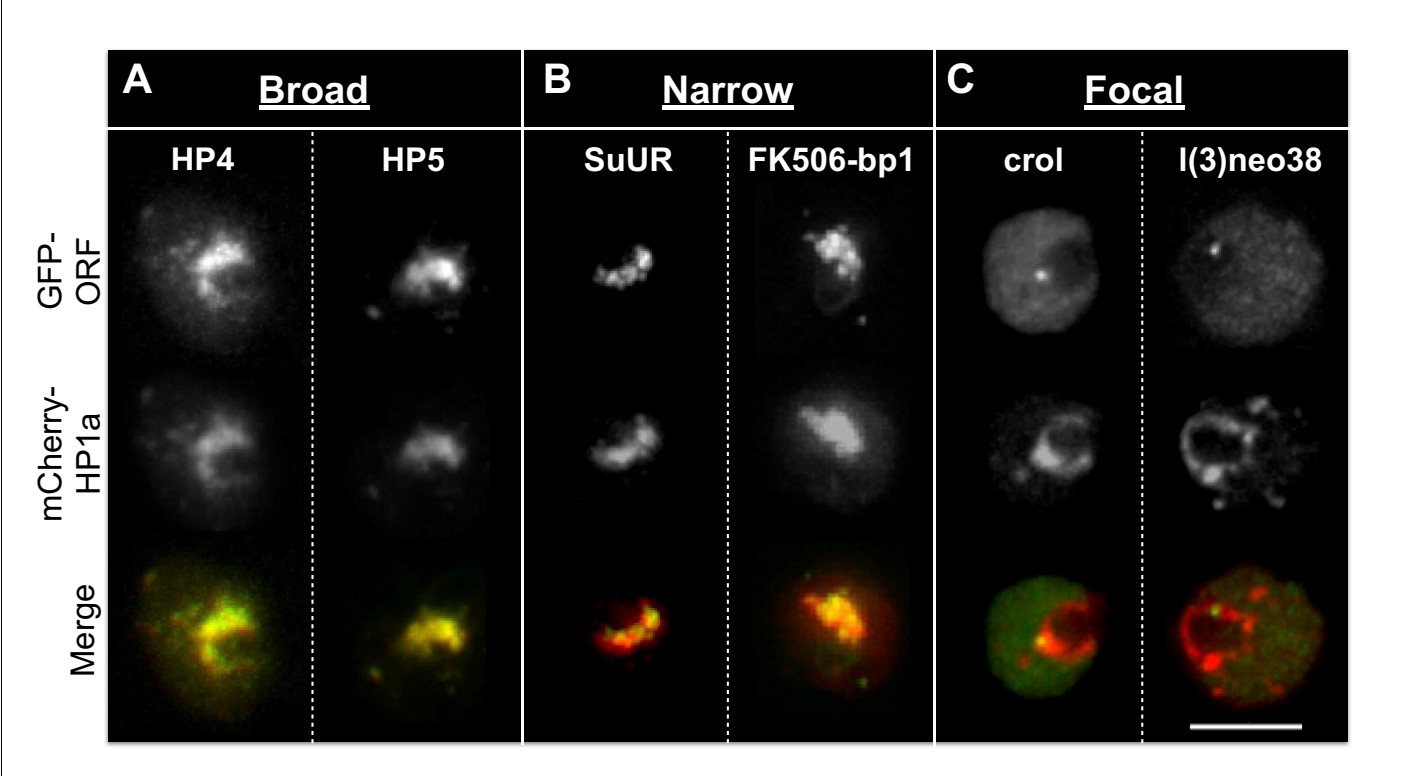

**Figure 5.** Heterochromatic proteins display diverse localization patterns. HP4 and HP5 broadly overlap with HP1a. SuUR and FK506-bp1 overlap with the interior of HP1a (narrow). Crol and l(3)neo38 form a focus within the HP1a domain (focal). Focal proteins are presented as slices, broad and narrow proteins are projections. mCherry-tagged HP1a is in red, GFP-tagged ORF is in green. Scale bar is 5 μm.

The following figure supplement is available for figure 5:

**Figure supplement 1.** Heterochromatic proteins display diverse localization patterns.

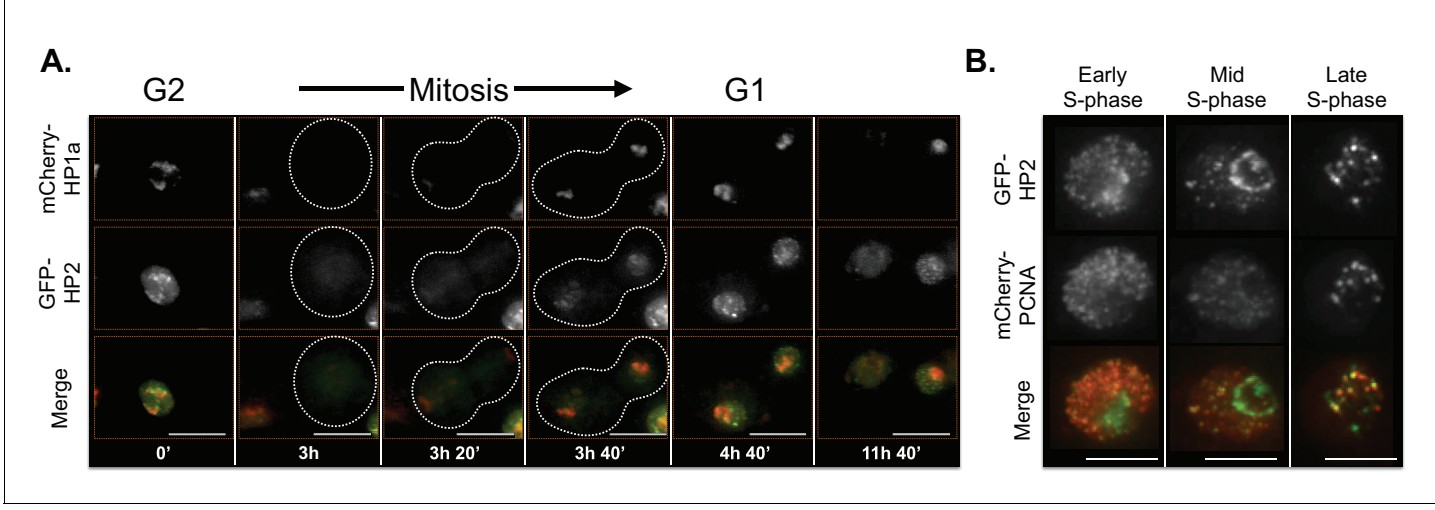

**Figure 6.** HP2 time-lapse imaging reveals dynamic regulation and overlap with PCNA throughout S-phase. HP2 partially overlaps and is enriched at the boundary of HP1a in G2, released from chromatin during mitosis and broadly colocalized with HP1a during G1. Mitosis is used to discriminate G1 from G2. Dotted lines indicate the cell periphery during mitosis. mCherry-tagged HP1a is in red, GFP-tagged HP2 is in green. Scale bar is 10 μm. (**B**) HP2 overlaps with PCNA foci in early, mid and late S-phase. Representative images of early, mid and late S-phase are shown. mCherry-tagged PCNA is in red, GFP-tagged HP2 is in green. Scale bar is 5 μm.

and inhibits fork progression (*Nordman et al., 2014*). Intriguingly, we observed a similar pattern for HP2 during S phase; euchromatic HP2 foci appear during S-phase (*Figure 6A*, 11h 40') and completely overlap with PCNA (*Figure 6B*). Determining if HP2 also functions during replication will require further investigation. Interestingly, HP2 and SuUR localization patterns and dynamics differ during G1 and G2, perhaps reflecting different biological roles during these cell cycle phases. Just prior to mitosis (in G2, *Figure 6A*, 0'), HP2 forms a narrow subdomain within the HP1a domain and is at the HP1a boundary in the same nucleus. During mitosis, HP2 is largely removed from chromosomes (*Figure 6A*, 3h) until anaphase/telophase, when HP2 is recruited shortly after the HP1a domain reforms (*Figure 6A*, 3h 20'). Then in G1, HP2 and HP1a broadly colocalize (*Figure 6A*, 3h 40') with the brightest HP2 signal at the HP1a domain boundary (*Figure 6A*, 4h 40'). In contrast, SuUR is not as dynamic as HP2; it forms a narrow subdomain within the HP1a domain during both G1 and G2, but is also released from heterochromatin during mitosis (*Figure 7—figure supplement 1*).

ADD1-PB and Odj also display different dynamic heterochromatin localization patterns. ADD1-PB forms bright focal subdomains with weaker broad enrichment in bulk heterochromatin, whereas Odj forms focal subdomains in G1 that broadly colocalizes with HP1a by the end of G2. A striking observation is that the intensities of both proteins are lower in G1 compared to G2, suggesting progressive heterochromatin loading of these proteins during interphase (*Figure 7—figure supplements 2* and *3*, and *Videos 3* and *4*). Interestingly, even though most

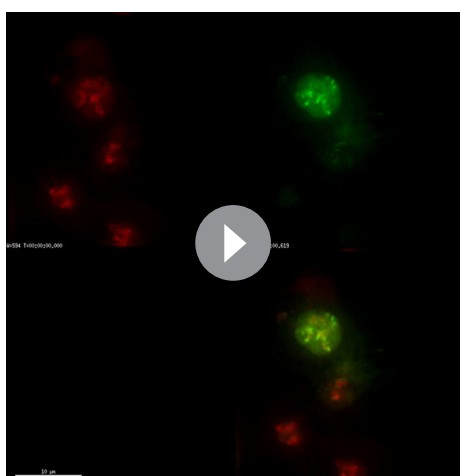

**Video 1.** HP2 time-lapse imaging reveals dynamic regulation throughout the cell cycle. HP2 partially overlaps and is enriched at the boundary of HP1a in G2, released from chromatin during mitosis and broadly colocalized with HP1a during G1. Mitosis is used to discriminate G1 from G2. mCherry-tagged HP1a is in red, GFP-tagged HP2 is in green. Scale bar is 10 μm.

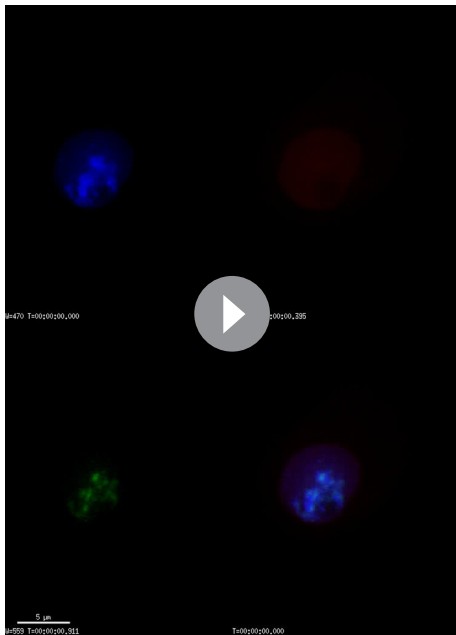

**Video 2.** Combined SuUR, HP1a and PCNA time-lapse imaging reveals dynamic regulation. SuUR colocalizes with HP1a during G2 and G1, and colocalizes with PCNA during S-phase. Mitosis is used to discriminate G1 from G2, while PCNA foci indicate S-phase. Cerulean-tagged HP1a is in blue, YFP-tagged SuUR is in green, mCherry-tagged PCNA is in red. Scale bar is 5 μm.

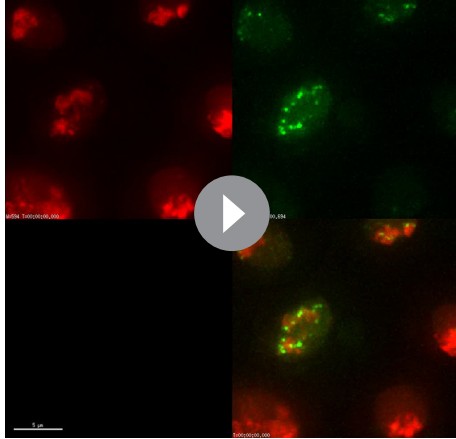

**Video 3.** ADD1-PB time-lapse imaging reveals dynamic regulation. ADD1-PB forms focal subdomains that abut and overlap HP1a, b not overlap with the centromeric or telomeric markers CID and HOAP (data not shown), respectively. In G2 ADD1-PB is predominantly focal at the heterochromatin boundary. A small amount of discrete signal remains on chromatin during mitosis and persists at low levels into G1, before eventually increasing in intensity, which suggests loading at the end of G1 or during S-phase. Mitosis is used to discriminate G1 from G2. mCherry-tagged HP1a is in red, GFP-tagged ADD1-PB is in green. Scale bar is 5 μm.

ADD1 and all detectable HP1a are removed during mitosis, some ADD1 signal remains attached to chromosomes, distinguishing it from all other HPips analyzed here. Another striking example is FK506-bp1, which displays a narrow localization pattern as well as a ring around the nucleolus throughout much of the cell cycle (*Figure 7—figure supplement 4* and *Video 5*). Interestingly, FK506-bp1 accumulates foci outside of heterochromatin during G2, which do not colocalize with markers for replication (PCNA), centromeres (CID) or telomeres (HOAP) (data not shown).

Supporting the validity of our approach, Lhr displayed its previously described localization pattern (*Brideau et al., 2006*) (broad and centromeric; *Table 3*, *Figure 7—figure supplement 5* and *Video 6*). However, in contrast to a previous report that XNP/ATRX is broadly enriched at polytene chromocenters (*Bassett et al., 2008*), we observe that XNP exhibits narrow and focal localization patterns in S2 cells (*Figure 7—figure supplement 6* and *Video 7*). This is consistent with XNP's observed enrichment at active genes, satellite DNA and heterochromatin of the X chromosome in imaginal discs and polytene chromosomes (*Schneiderman et al., 2009*; *Emelyanov et al., 2010*). Emphasizing the complexity of subdomain architecture, we detected some Lhr and XNP foci that colocalize within the same nucleus, while others do not (*Figure 7—figure supplement 5C*).

We conclude that the localization patterns for 5 of the 7 HPips studied with time-lapse imaging are dynamic throughout the cell cycle (*Figure 7*). Further analysis is required to determine if the changing distributions throughout the cell cycle reflects biological functions. For instance, the prevalence of ADD1-PB foci localized at the heterochromatin boundary could indicate a role in maintaining the border between the heterochromatin and euchromatin domains.

## Discussion

The heterochromatin domain is defined molecularly by enrichment for HP1, which binds many different proteins and has been implicated in diverse and sometimes contradictory functions, including repression of transposons (*Lundberg et al., 2013*) and genes, and promotion of gene expression

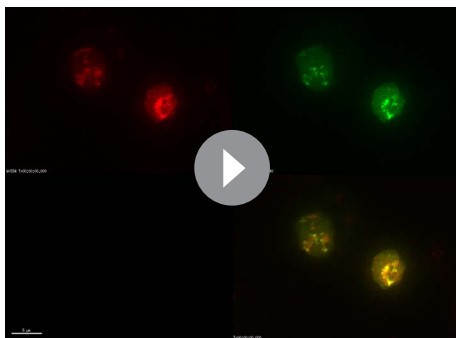

**Video 4.** Oddjob time-lapse imaging reveals dynamic regulation. Odj broadly co-localizes with HP1a at the end of G2 and disperses from chromosomes during mitosis. It reforms as a focal subdomain after mitosis that gradually increases in size, until it broadly overlaps HP1a again. Mitosis is used to discriminate G1 from G2. mCherry-tagged HP1a is in red, GFP-tagged Odj is in green. Scale bar is 5 μm.

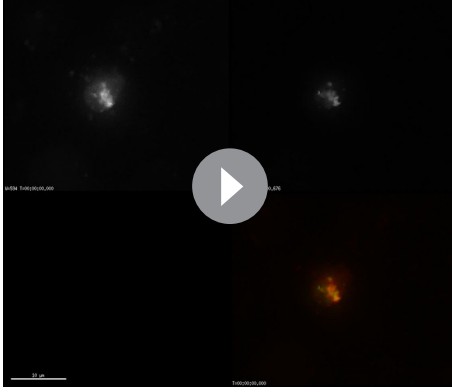

**Video 5.** FK506-bp1 time-lapse imaging reveals dynamic regulation. FK506-bp1 narrowly co-localizes with HP1a throughout much of the cell cycle and loses co-localization with HP1a 20 min to 1 hr before HP1a is released from chromosomes (prophase). After mitosis, the narrow co-localization pattern of FK506-bp1 is restored, with a weak ring around the nucleolus, which is located adjacent to the HP1a domain. FK506-bp1 foci then begin to accumulate outside of heterochromatin until just before prophase, when they disappear prior to HP1a removal. Foci do not track with PCNA (replication), CID (centromeres) or HOAP (telomeres) foci (data not shown). Mitosis is used to discriminate G1 from G2. mCherry-tagged HP1a is in red, GFP-tagged FK506-bp1 is in green. Scale bar is 10 μm.

(*Piacentini et al., 2009*). How HP1a mediates a wide variety of heterochromatin functions and maintains interactions with multiple binding partners is currently unknown. To better understand this important nuclear domain, we performed two complementary screens to identify novel structural and functional components of *Drosophila* heterochromatin. We immunoprecipitated HP1a under stringent conditions and performed LC-MS/MS to identify core heterochromatin components. In addition, an unbiased genome-wide RNAi screen was used to identify regulators of heterochromatin organization, maintenance and establishment, independent of their ability to bind HP1a. These screens identified 118 novel putative HP1a interactors and 374 putative regulators of HP1a. Candidate hits were further analyzed for heterochromatin localization, and 34% (30/89 tested) strongly colocalized with HP1a in low-resolution imaging. Higher-resolution microscopy and time-lapse analysis revealed that many heterochromatin proteins occupy subdomains within the holodomain, and display dynamic localization patterns throughout the cell cycle. We identified at least 13 previously unknown components required for heterochromatin-mediated gene silencing (PEV), and demonstrated that the organization and composition of heterochromatin is more complex and dynamic than suggested by previous studies (see *Supplementary file 5* for a summary of the results from our study).

Although IP-MS was previously used to identify HP1a interacting proteins (*Rosnoblet et al., 2011*; *Lechner et al., 2005*; *Motamedi et al., 2008*; *Ryu et al., 2014*; *Alekseyenko et al., 2014*), our results demonstrate that many new heterochromatin components can still be isolated using this approach. Here, we optimized purification conditions by expressing tagged-HP1a at low levels, using high salt concentrations and removing DNA and RNA, which may have released different subpopulations of HP1a complexes than previous studies. Regardless, this approach was validated by identifying at least 17 previously known pericentromeric heterochromatin structural components (e.g. HP2, HP3/Lhr, HP4, HP5, Su(var)3–9, *Table 1—source data 1* and *2*). Most importantly, we isolated 118 HPips that were not previously associated with heterochromatin. The relevance of these novel HPips to heterochromatin structure and function was demonstrated by cytological and phenotypic analyses. First, 2 of these novel HPips (FK506-bp1 and Odj) colocalized with HP1a using high-resolution imaging. Second, mutations in 3 of 5 novel HPips tested (Hsc70-4, fl(2)d and Odj) act as Su(var)s (*Figure 4* and *Figure 4—source data 1*), demonstrating relevance to transcriptional silencing, a well-established heterochromatin function.

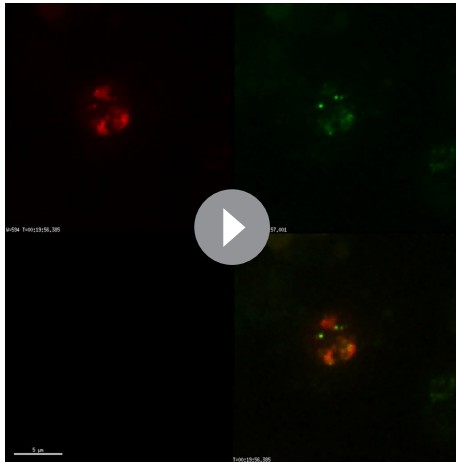

**Video 6.** Lhr time-lapse imaging reveals dynamic regulation. Lhr broadly co-localizes with HP1a and is released from chromatin during mitosis. Mitosis is used to discriminate G1 from G2. mCherry-tagged HP1a is in red, GFP-tagged Lhr is in green. Scale bar is 5 μm.

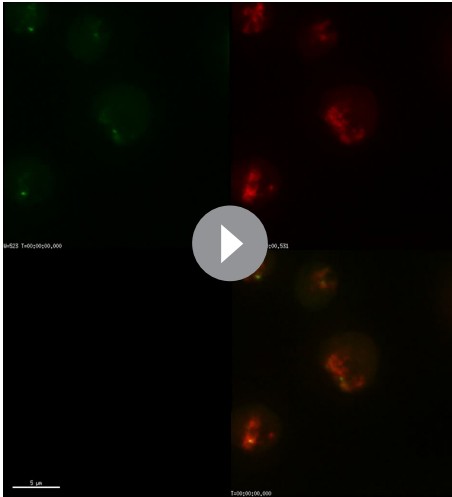

**Video 7.** XNP time-lapse imaging reveals dynamic regulation. XNP colocalizes with a portion of HP1a in G2. The majority of XNP is removed during mitosis, however 1-2 foci remain chromatin-bound. In G1 XNP is focal within the HP1a domain but gradually accumulates in size and colocalizes with more HP1a. Mitosis is used to discriminate G1 from G2. mCherry-tagged HP1a is in red, GFP-tagged XNP is in green. Scale bar is 5 μm.

In contrast, a genome-wide RNAi screen for regulators of heterochromatin architecture has not been reported previously. We utilized multiple methods to identify candidates that disrupted HP1a levels or localization, including ranking gene depletions by changes in HP1a intensity, as well as supervised clustering (trained on results from HP1a depletions) and unsupervised clustering (hits selected based on similarity to HP1a or Su(var)3–9 depletions). All three methods identified known regulators of heterochromatin (*Table 2*), yet there was surprisingly little overlap between the different approaches. This suggests that more than one method of hit identification may be needed for high-content screening of potentially subtle cytological phenotypes. Regardless, we identified 374 candidate regulators whose loss mimicked HP1a depletion, including 355 with no previously known connection to heterochromatin. Importantly, 7 of 12 tested candidates (58%) colocalized with HP1a at high-resolution (*Table 3*), and mutations in 9 of 10 tested candidates acted as suppressors of PEV (*Figure 4* and *Figure 4—source data 1*).

It is important to note that 564 gene disruptions led to increased HP1a intensity; future analyses of these interesting candidates may reveal new factors that inhibit heterochromatin formation and/ or are required for removal of HP1a. Additionally, since we extracted 32 imaging features per nucleus and imaged ~400 nuclei per gene depletion, further mining of this extremely rich dataset, in addition to cytological and phenotypic analyses of the remaining candidates, are likely to identify additional proteins that impact heterochromatin, as well as regulators of other processes (e.g. apoptosis and the cell cycle). We conclude that the RNAi screen successfully identified novel heterochromatin components and regulators.

Interestingly, only HP1a and three other proteins were identified in both the RNAi and the HP1a IP-MS screens. A potential reason for poor overlap is that the RNAi screen enriched for hits upstream of HP1a deposition, while the IP-MS enriched for genes acting downstream of HP1a loading onto chromatin. This hypothesis is supported by studies demonstrating that loss of HPips does not cause visible defects in HP1a domain organization (e.g. SMC5/6 [*Chiolo et al., 2011*], KDM4A [Colmenares et al., unpublished]). Therefore, we propose that the complementary approaches utilized in this study enabled identification of different classes of heterochromatin proteins. Finally, the majority of heterochromatin-localized hits are required for transcriptional silencing (8/11 = 73%), suggesting that further analysis of the RNAi screen hits will identify more heterochromatin regulators.

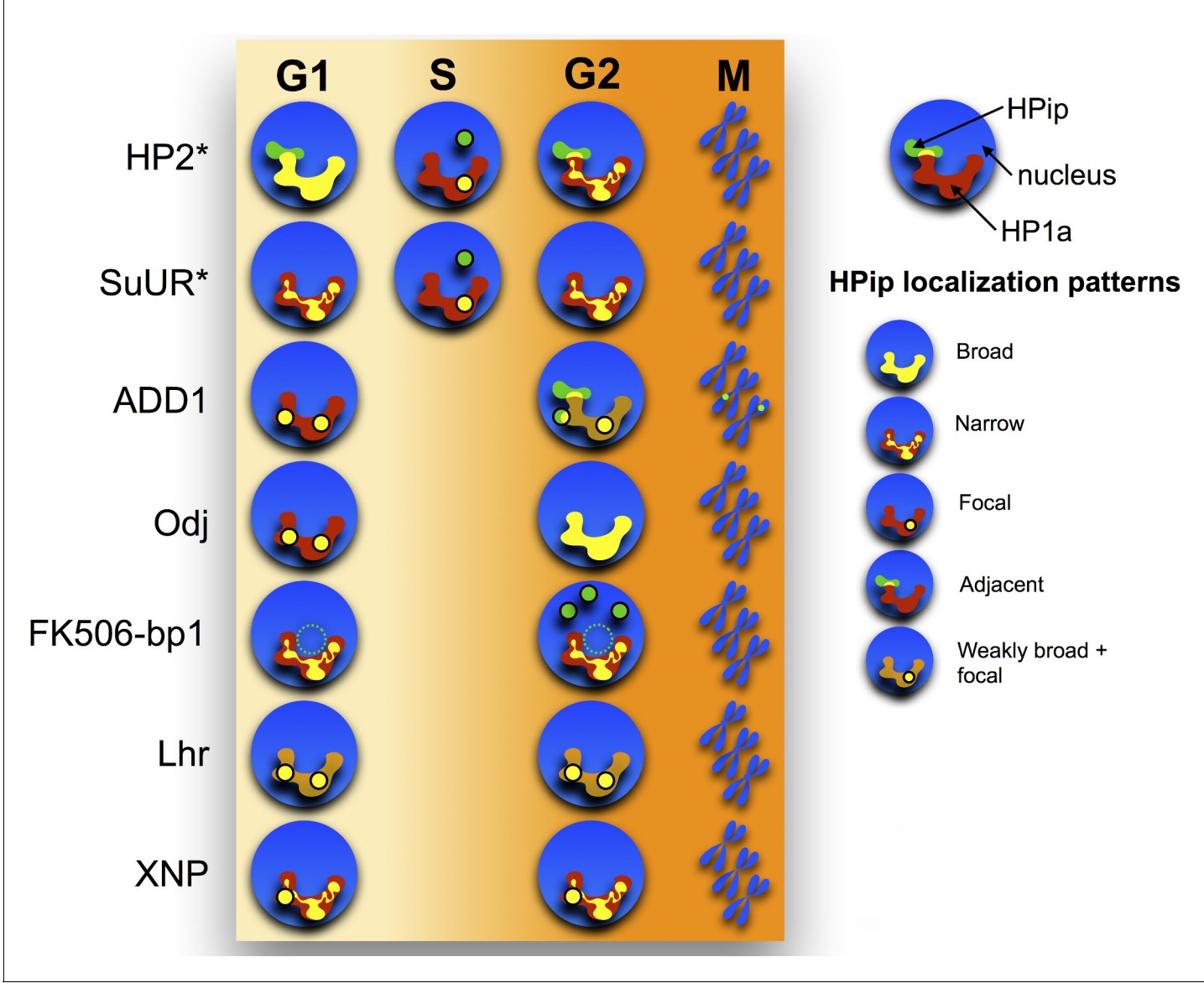

**Figure 7.** Time-lapse imaging reveals a variety of dynamic localization patterns within heterochromatin. A graphical representation of the localization patterns of heterochromatic proteins throughout the cell cycle is shown. HP1a is depicted in red, the heterochromatin protein (HPip) in green and overlap between the two in yellow. A dotted circle indicates that FK506-bp1 forms a ring around the nucleolus. * indicates foci overlap completely with PCNA during S-phase.

The following figure supplements are available for figure 7:

**Figure supplement 1.** Combined SuUR, HP1a and PCNA time-lapse imaging reveals dynamic regulation.

**Figure supplement 2.** ADD1-PB time-lapse imaging reveals dynamic regulation.

**Figure supplement 3.** Oddjob time-lapse imaging reveals dynamic regulation.

**Figure supplement 4.** FK506-bp1 time-lapse imaging reveals dynamic regulation.

**Figure supplement 5.** Lhr time-lapse imaging reveals dynamic regulation.

**Figure supplement 6.** XNP time-lapse imaging reveals dynamic regulation.

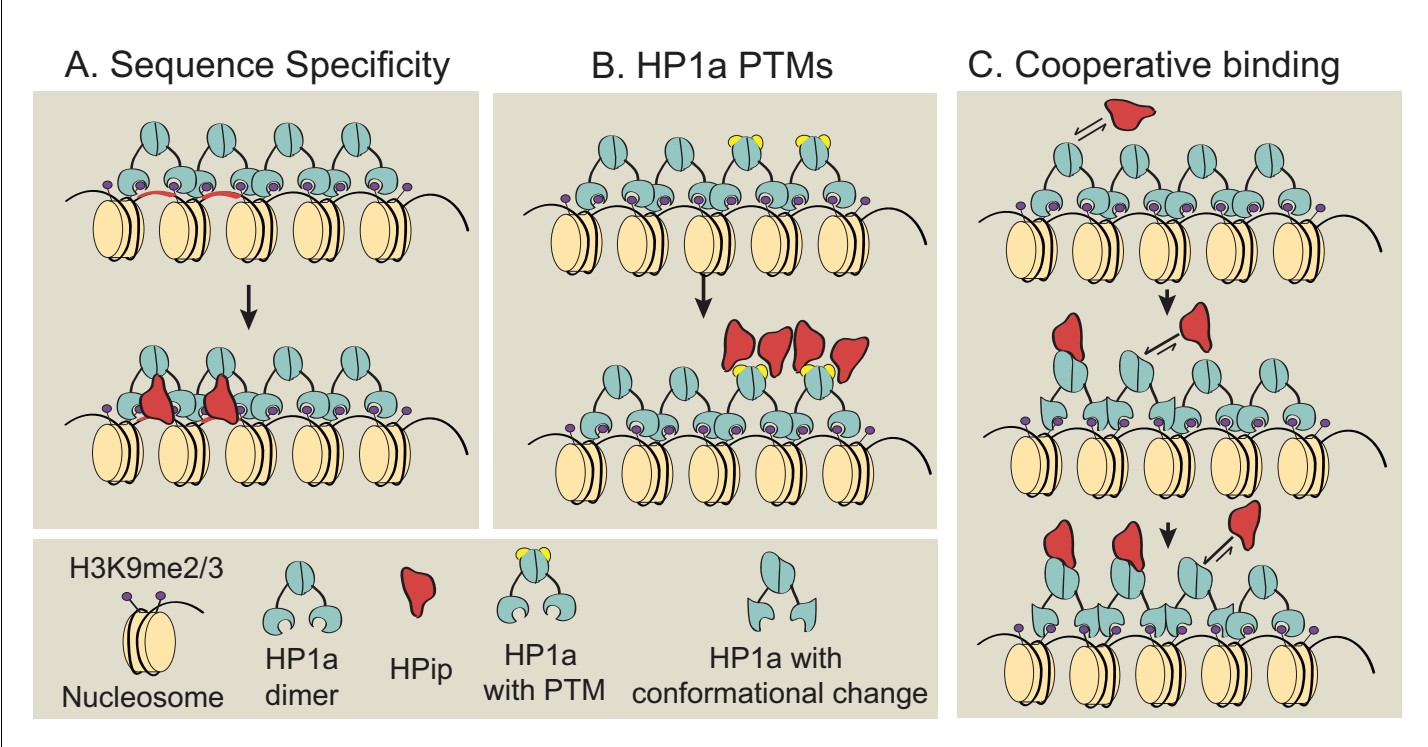

**Figure 8.** Models for subdomain formation within heterochromatin. We propose three non-mutually exclusive models for subdomain formation of HP1a interacting proteins (HPips) within the HP1a (teal) heterochromatin holodomain. (**A**) The HPip (red) may be recruited to a specific sequence and seeds the formation of a subdomain (as observed for D1 [**Aulner et al., 2002**] and GAGA [**Raff et al., 1994**] factor). (**B**) HP1a and its orthologs are extensively post-translationally modified by SUMOylation, acetylation, methylation, formylation, ubiquitination and poly(ADP-ribosyl)ation (**Alekseyenko et al., 2014**; **Lomberk et al., 2006**; **LeRoy et al., 2009**). An HPip could have an increased binding affinity for a specific HP1a PTM (yellow). Thus, HP1a PTMs may regulate HP1a complex formation and spatially restrict HPip recruitment. Consistent with the PTM model, HP2 and PIWI have been shown to have higher binding affinities for HP1a proteins containing phospho-mimic mutations in the HP1a chromo shadow domain (**Mendez et al., 2011**). (**C**) Subdomains could form by a cooperative binding mechanism (**Bray and Duke, 2004**; **Bai et al., 2010**). HP1a can oligomerize at least up to tetramers (**Wang et al., 2000**; **Zhao et al., 2000**; **Canzio et al., 2011**), forming a multivalent platform for HPip binding (i.e. more than one HPip binding site per HP1a oligomer). Thus, initial binding by an HPip could induce a higher binding affinity between a neighboring HP1a molecule and the HPip. The dotted arrow indicates potential self-interactions between HPips and solid arrows indicate hypothetical HPip on/off rates.

It is important to note that the absence of either HP1a colocalization or transcriptional silencing effects does not eliminate candidates from having important roles in heterochromatin structure or function. First, proteins that localize to euchromatin can regulate heterochromatin; for example, the euchromatic JIL-1 kinase restricts heterochromatin spreading, and JIL-1 mutants modify silencing phenotypes (**Zhang et al., 2006**). Second, the modification of silencing assays only one of many known heterochromatin properties and functions (**Bernard et al., 2001**; **Dernburg et al., 1996**; **Karpen et al., 1996**; **McKee and Karpen, 1990**; **Peng and Karpen, 2009**; **Sienski et al., 2012**; **Clowney et al., 2012**). Third, heterochromatin proteins that form subdomains may only affect PEV of genes inserted in their local environment. Thus, to exhaustively identify proteins involved in heterochromatin structure and function, more of the candidates identified in our screens need to be analyzed for colocalization with HP1a, and for impact on other heterochromatin functions, such as DNA repair (**Chiolo et al., 2011**) and chromosome segregation (**Dernburg et al., 1996**; **Karpen et al., 1996**; **McKee and Karpen, 1990**). Furthermore, inclusion of reporters located in other chromosomes, in addition to the Y chromosome PEV reporter utilized here, will determine if subdomain proteins exert local versus widespread PEV effects.

In addition to identifying novel heterochromatin components and regulators, analysis of localization patterns revealed that heterochromatin organization is complex and dynamic. Previous studies using polytenized chromosomes showed that the majority of HPips are broadly distributed across the heterochromatin domain, and that a few heterochromatin proteins localized to sub-regions

within heterochromatin (e.g. piwi [*Brower-Toland et al., 2007*], ATF-2 [*Seong et al., 2011*]), in some cases due to binding to specific repeated DNAs (e.g. prod [*Platero et al., 1998*], D1 [*Aulner et al., 2002*]). However, using cells without polytene chromosomes, we showed that the majority of heterochromatin proteins analyzed (17 of 22) form subdomains within heterochromatin, and that 5 of the 7 proteins analyzed by live imaging display highly dynamic localization patterns throughout the cell cycle. Importantly, the localization patterns for GFP-tagged HPips recapitulated published results (e.g. HP4, HP5, Lhr) (*Greil et al., 2007*), suggesting that GFP-tagging per se was not responsible for the diverse patterns observed here. Additionally, we showed that proteins that localize broadly to the underreplicated heterochromatin in non-cycling nuclei with polytenized chromosomes (e.g. SuUR [*Makunin et al., 2002*], HP2 [*Shaffer et al., 2002*]) can form subdomains in S2 cells.

Previous studies showed that many but not all HP1a binding partners contain a conserved PxVxL-like motif (*Smothers and Henikoff, 2000*; *Nozawa et al., 2010*; *Huang et al., 2006*), and that the HP1a chromo shadow domain and C-terminal extension, as well as residues near the PxVxL, determine the strength of HP1a:HPip interactions (*Mendez et al., 2011*). However, the prevalence of subdomain localization patterns within heterochromatin demonstrates that binding to HP1a is not indiscriminate, and must require other, currently unknown mechanisms. We consider three possibilities for subdomain formation that are not mutually exclusive (*Figure 8*): 1) sequence-specific binding, 2) binding to specific HP1a posttranslational modifications, and 3) cooperative binding between HP1a and a HPip. Further studies are required to determine if these or other mechanisms are responsible for establishing or maintaining the specific and diverse localization patterns observed for HPips and other heterochromatin proteins. One key question is whether subdomains form at the same genomic locations in every cell, or are initiated and grown at random genomic sites.

This study reveals unexpected complexity within heterochromatin, in terms of both the number of new structural and functional components identified, and their localization to discrete, dynamic subdomains. We speculate that broadly distributed proteins could encode structural components important for universal aspects of heterochromatin architecture and function (e.g. nucleosomal ordering, variable accessibility of exogenous proteins, and coalescence of heterochromatin domains; reviewed in [*Elgin and Reuter, 2013*]). In contrast, subdomains may regulate specific functions or localized, dynamic structural changes (e.g. decreased compaction to enable transcription, histone exchange). Heterochromatin may be just as structurally and functionally dynamic and diverse as euchromatin, and increased understanding of its organization will likely yield important insights into the nuclear architecture and genome biology. Thus, it will be important to determine the mechanisms responsible for subdomain formation, and how they contribute to specific heterochromatin functions.

## Materials and methods

### Single-step immunopurification

$2 \times 10^9$ S2 cells stably expressing FS-HP1a (3X-FLAG-Myc-StrepII-PP-HP1a [where PP is a PreScission Protease cut site]) under control of the *copia* promoter (plasmid construction as described in [*Chiolo et al., 2011*]) were exposed or mock-exposed to 10 Gy of X-rays using a 160 kV source. Cells were allowed to recover 10 or 60 min, harvested at 600 r.c.f for 5 min and flash frozen in liquid nitrogen prior to resuspension/lysis in Buffer A (0.05% NP-40, 50 mM Hepes pH 7.6, 10 mM KCl, 3 mM MgCl$_2$, 10% glycerol, 5 mM NaF, 5 mM β-glycerophosphate, 1 mM Benzamidine, 1X protease inhibitor cocktail [Roche, Basel, Switzerland: 11 836 170 001], 1 mM PMSF, 25mM NEM, 1:1000 Phosphatase Inhibitor Cocktail 2 [Sigma-Aldrich, St. Louis, MO: P5726], 1:1000 Phosphatase Inhibitor Cocktail 3 [Sigma P0044], 1:1000 Protease Inhibitor Cocktail [Sigma P8340]) in 500 μl/2 × 10$^8$ cells. Cell extracts were treated with 10 units benzonase (EMD Millipore, Hayward, CA: 80601–766) per 37 μg of chromatin (estimated by A$_{260}$ reading) at 4°C with mixing for 30 min. Nuclease digest was stopped with 0.5 mM EDTA, and HP1a was extracted on ice with 300mM NaOAc for 1 hr with mixing. Cell extracts were cleared by centrifugation at 16,100 r.c.f. for 10 min at 4°C. Supernatant was transferred to a new tube and mixed with anti-3XFLAG M2 beads (Sigma) O/N at 4°C. Bound material was washed four times with Buffer A at 4°C while mixing, eluted with 3XFLAG peptide and concentrated using Amicon Ultra-0.5 Centrifugal Filter Unit with Ultracel-3 membrane. LC-MS/MS was performed at Zentrallabor für Proteinanalytik (Protein analysis Unit, Medical School of Ludwig-

Maximilians University of Munich, Germany). Protein material similarly isolated from S2 cells lacking FS-HP1a expression were also analyzed as a negative control.

## Tandem affinity immunopurification

Tandem affinity immunopurification was performed essentially as described for single-step immunopurification except FS-HP1a was incubated with anti-3XFLAG M2 beads (Sigma) for 2 hr at 4°C and then bound to Strep-Tactin Superflow beads (IBA, Goettingen, Germany) O/N at 4°C and washed and eluted per manufacturer's instructions. LC-MS/MS was performed at the Scripps Center for Metabolomics and Mass Spectrometry.

## Database searching

All MS/MS samples were analyzed using Mascot (Matrix Science, London, UK; version 2.3.02) and X! Tandem (The GPM, thegpm.org; version CYCLONE (2010.12.01.1)). Mascot was set up to search the *Drosophila* NCBI protein database (downloaded 2010; 14,335 entries). X! Tandem was set up to search a subset of the *Drosophila* NCBI protein database assuming the digestion enzyme trypsin. Mascot and X! Tandem were searched with a fragment ion mass tolerance of 0.50 Da and a parent ion tolerance of 10.0 PPM for single-step immunopurification. Mascot and X! Tandem were searched with a fragment ion mass tolerance of 0.80 Da and a parent ion tolerance of 2.0 Da for tandem-step immunopurification. Iodoacetamide derivative of cysteine was specified in Mascot and X! Tandem as a fixed modification. Methylation of lysine, oxidation of methionine and phosphorylation of serine, threonine and tyrosine were specified in X! Tandem as variable modifications. Methylation of lysine, oxidation of methionine, acetaldehyde +28 of lysine, formylation of lysine, acetylation of lysine, tri-methylation and di-methylation of lysine and phosphorylation of serine, threonine and tyrosine were specified in Mascot as variable modifications. Variable modifications were accepted if they could be established at greater than 95.0% probability by Mascot.

## Criteria for protein identification

Scaffold (version Scaffold_4.0.7, Proteome Software Inc., Portland, OR) was used to validate MS/MS-based peptide and protein identifications. Peptide identifications were accepted if they could be established at greater than 95.0% probability by the Peptide Prophet algorithm (*Keller et al., 2002*). Protein identifications were accepted if they could be established at greater than 95.0% probability and contained at least 2 identified peptides. Protein probabilities were assigned by the Protein Prophet algorithm (*Nesvizhskii et al., 2003*). Proteins that contained similar peptides and could not be differentiated based on MS/MS analysis alone were grouped to satisfy the principles of parsimony.

## Genome-wide RNAi screen

10 µL of logarithmically growing *Drosophila melanogaster* Kc embryonic tissue culture cells were seeded at a density of $1 \times 10^6$ cells/mL in serum-free Schneider's medium (Invitrogen, Carlsbad, CA) on 384-well plates (Corning, Corning, NY: #3712) containing 0.25 µg dsRNA per well. Cells were incubated with dsRNA at room temperature for 30 min. 30 µl of Schneider's medium (Invitrogen) with $1\times$ antibiotics (Invitrogen), and 10% FCS was added to each well. Plates were incubated for 4 days at 25°C in a humid chamber. Cells were exposed to 5 Gy of X-rays using a Faxitron TRX5200 operated at 130 kV and allowed to recover for 60' prior to fixation (the results of the radiation aspect of the screen are not reported here). Cells were fixed for 5 min with 3.7% paraformaldehyde and washed 3X for 5 min in PBS with 0.5% Triton X-100 (PBST). Cells were treated for 30 min with blocking solution (PBST containing 5% FCS), followed by overnight 4°C incubation in 10 µl of blocking solution containing 1:500 mouse anti-HP1a antibody (Developmental Studies Hybridoma Bank, University of Iowa, Iowa City, Iowa: C1A9c) and 1:1000 rabbit anti-γH2Av (Rockland/VWR, Limerick, PA: VWR #600-401-914). Cells were then washed 3X for 5 min with PBST, incubated with 10 µl of blocking solution containing secondary antibodies (Alexa 488-conjugated anti-mouse and Alexa 546-conjugated anti-rabbit antibody at 1:500 dilutions [Invitrogen]) for 1 hr at room temperature, washed 2X with PBST and 1X with PBS. DNA was stained with 10 µl of 0.2 µg/ml of DAPI in PBS for 5 min at room temperature and washed with PBS. Cell plating was performed using a CombiDrop and IF protocol was performed using a V11 Bravo at the Berkeley Screening Center. Plates were imaged using

a Zeiss Axio Observer Z1 automated microscope (Carl Zeiss, Jena, Germany), with a Zeiss EC Plan-Neoflaur 40X objective (N.A. of 0.75).

## Image analysis for genome-wide RNAi screen

All image manipulations and analyses were done with Matlab (MathWorks, Inc., Natick, MA) and DIPimage (image processing toolbox for Matlab, Delft University of Technology, The Netherlands). The Matlab code is available at https://github.com/svcostes/Elife_Pearson_Script. Nuclear segmentation was performed as previously described (*Costes et al., 2006*). Briefly, background heterogeneity was corrected by subtracting the original image blurred by a Gaussian filter of the appropriate size. A constant threshold was then used to identify all nuclei independently of their varying intensities. Touching nuclei were separated using watershed approaches. Briefly, the distance transform of a binary mask encompassing more than one nucleus typically yields multiple bright spots representing the center of each nucleus. These maxima were used as seeds and expanded to fill the binary mask, allowing the separation of each individual nucleus. We used the DIPimage object measurement function to extract a large array of imaging features for DAPI, HP1a and γH2Av intensity, as well as pairwise correlations (*Costes et al., 2004*) between HP1a, γH2Av and DAPI. The nuclei were numbered and their boundaries demarcated on a large field of view to enable visual verification of the automatically generated data set.

## Data normalization and statistics for genome-wide RNAi screen

Data processing was carried out in the R Environment (*R Core Team, 2013*), and Rank Product analysis (*Breitling et al., 2004*) was performed using the Bioconductor package (*Gentleman et al., 2004*) to normalize the data and obtain a p-value estimation (with 100 permutations used to calculate the null density and subsequent p-value estimation). Note that all hits identified below were manually inspected to ensure that the images were in focus.

## Identification of genome-wide RNAi screen hits using HP1a intensity

Rank Product estimated p-value cut-offs to identify hits for decreased relative HP1a maximum (i.e. maximum/mean), increased HP1a Kurtosis, decreased HP1a average and decreased HP1a maximum intensity (collectively 'HP1a metrics') were chosen based on maximal inclusion of HP1a positive controls, and correspond to 1.5E-03, 5.0E-04, 6.3E-04 and 3.8E-04, respectively. Hits displaying increased cell death were eliminated based on the nuclear morphology and the number of nuclei per field. Hits displaying increased HP1a intensity were chosen by taking the overlap of hits with increased HP1a mean intensity (p-value<0.05) and increased HP1a maximum intensity (p-value<0.05), and discarding hits that lead to a decreased cell number (p-value<0.05).

## Identification of genome-wide RNAi screen hits using support vector machines

Two SVMs, using polynomial kernels, were trained based on positive controls (HP1a RNAi) and negative controls (mock RNAi, GFP RNAi, Rho1 RNAi [produces binucleate cells], Thread RNAi [induces cell death]) using Rank Product ranks. The classifier was then applied to the entire dataset and the identification of HP1a knockdowns withheld from the training set was used to optimize the SVM. The SVM utilized either all imaging features or all imaging features except those associated with γH2Av (denoted "SVM - HP1a only features" in *Figure 2—source data 1*). SVM analysis was performed using the R package svmpath (*Hastie et al., 2004*) with a ridge value of 1E-08 and a kernel parameter of 0.8 for all imaging features, or 0.4 for HP1a features.

## Identification of genome-wide RNAi screen hits using hierarchical clustering

Rankings from Rank Product analysis using all imaging features or HP1a only imaging features were used to calculate (using Matlab) the pair-wise distance between every sample using multiple distance measures (Spearman, Mahalanobis, Euclidean and Pearson). The data were then randomized and the distances of the randomized data were measured repeatedly. An estimated p-value was derived by specifying that the average distance found at the 1 percentile corresponded to a p-value of 0.01. We used a p-value cut-off of 5E-07 to determine significant distances from HP1a or Su(var)3–9 RNAi-

treated cells. Next, we identified genes that were pair-wise close to at least five HP1a RNAi-treated samples by more than one distance metric. Finally, Matlab's dendrogram function was applied to the HP1a Pearson correlation coefficient distance matrix and used to cluster the data. Hits clustering with HP1a were visually identified using Matlab's clustergram function.

## Gene ontology enrichment analysis

Database for Annotation, Visualization and Integrated Discovery (DAVID) v6.7 (*Huang et al., 2008*, *2009*) was used to identify enriched GO terms. Functionally similar annotations as determined by Annotation Clustering in DAVID were not reported unless otherwise indicated.

## Plasmid generation

The pCopia-LAP-loxP acceptor plasmids were obtained by insertion of PCR-amplified loxP site, pro-karyotic promoter and splice acceptor from pMK33-CTAP with AscI and PacI overhangs into AscI/PacI digested pCopia-LAP (*Cheeseman and Desai, 2005*). BS clones were subcloned into pCopia-LAP-loxP plasmids as in (*Yu et al., 2011*) and named pCopia-LAP-loxP-ORF-loxP. pCopia-LAP-loxP-ORF-loxP plasmids were used for the high-throughput low-resolution screen using InCell6000 imaging. All other live imaging was done using pCopia-LAP-ORF or pCopia-ORF-LAP plasmids. pCopia-ORF-LAP was generated by removal of the 5' LAP tag from pCopia-LAP-ORF and introduction of a 19 amino acid polylinker using Gibson cloning and LAP tag 3' of an ORF insertion site. ORFs were PCR-amplified from pCopia-LAP-loxP-ORF-loxP plasmids and cloned into AscI/PacI digested pCopia-LAP-ORF, or XbaI/PacI or NheI/PacI digested pCopia-ORF-LAP. Primers are listed in *Supplementary file 6*.

## Low-resolution imaging screen for HP1a colocalizing proteins

S2 cells were transiently transfected with pCopia-mCherry-HP1a and pCopia-GFP-loxP-ORF-loxP using TransIT-2020 (MIR 5400; Mirus Bio, Madison, WI). Cells were imaged 3 days post-transfection using an InCell 6000 (GE healthcare Bio-Sciences, Pittsburgh, PA, USA) in open aperture mode. We captured a single z-slice in 9 fields/well with a 20X-objective (0.75 NA). Nuclei were segmented as previously described (*Costes et al., 2006*) using mCherry-HP1a and selected for roundness using a metric based on the perimeter square over the area. Nuclei with average intensity in background range (for GFP <4,000 AU, for mCherry <3,000 AU) were discarded. Nuclei whose average intensity saturated the 16-bit camera were also discarded (less than 0.01% of nuclei), leaving ~200 nuclei on average/well (wells with less than 10 nuclei were discarded). The Pearson correlation coefficient (PCC) was calculated per nucleus. To determine the significance of the correlation between mCherry-HP1a and GFP-ORF, we compared the PCC of GFP-ORF and mCherry-HP1a to the PCC of GFP-modulo and mCherry-HP1a using a two-sided unpaired Mann-Whitney test. If a construct was transfected in duplicate then the highest scoring well was used.

## PEV assay

Top scoring proteins from the colocalization screen were assayed for silencing effects if they were previously unknown to modify PEV, and if fly mutant alleles or RNAi lines were available and geno-types did not preclude scoring *white* variegation (i.e. constructs not marked with *white+*; see *Figure 4* and *Figure 4—source data 1* for list of fly stocks). Mutant and RNAi fly stocks were all obtained from the Bloomington Stock center, except for Ago2[51B] which was a kind gift from F.B. Gao (*Xu et al., 2004*). Flies were first crossed into a *y, w* background with appropriate balancers, then females containing mutations were then crossed with *y, w, KV108* males. All stocks used are listed in *Figure 4—source data 1*. The KV108 line contains a SUPor-P construct with *y+* and *w+* reporter genes inserted in the heterochromatin of the Y chromosome, resulting in variegating eye and abdomen pigmentation (*Konev et al., 2003*). Female RNAi flies were crossed with *y, w, KV108* males harboring Act::GAL4. Adult male progeny from these crosses were aged 3–5 days, frozen and imaged for either *white* variegation in eyes or *yellow* variegation in the abdomen. Imaging was conducted on homozygous mutants when viable, otherwise heterozygous mutants were imaged. We detected very strong PEV suppression by TM3 balancer chromosomes, and therefore imaged only heterozygous mutants lacking this balancer. Mutant effects on PEV were compared with wildtype flies in a *y, w* background, whereas RNAi fly effects were compared with a mCherry RNAi fly stock.

To quantify *white* variegation, Color Inspector 3D (Kai Uwe Barthel, Berlin, Germany) in Fiji (*Schindelin, 2012*; *Schneider et al., 2012*, *2015*) was used to determine the RGB values of 'red' pixels (indicating loss of suppression) (0–255, 0–90, 0–20). The definition of 'red' was used uniformly across all samples to create a binary mask of the 'red' pixels in each eye. The area of the eye composed of 'red' pixels was then calculated (*Figure 4A*). The p-values were calculated with a 2-tailed, 2-sample unequal variance t-test using appropriate negative controls for each group (*Figure 4* and *Figure 4—source data 1*). Code is available at https://github.com/jmswenson/variegation.

Yellow variegation was quantified, in a double-blind manner, by manually counting the number of dark spots (i.e. where yellow is expressed) on the abdomen, and a p-value was calculated with the two-sample Kolmogorov-Smirnov test.

### High-resolution imaging and analysis

Images were taken using a 60X oil immersion objective (NA 1.40) on a Deltavision Spectris microscope (GE Healthcare) and images were deconvolved using SoftWoRx (Applied Precision, LLC). Time-lapse images were acquired once every 15–20 min. BioTAP-tagged ADD1 was colocalized with HP1a by performing IF with rabbit anti-peroxidase antibody (Sigma P1291) (1:100) and mouse anti-HP1a antibody (C1A9; Developmental Studies Hybridoma Bank) (1:500) in fixed S2 cells. Cells were fixed (4% PFA for 5 min) three days after transient transfection (TransIT-2020 MIR 5400; MirusBio). Slides were blocked in 1% milk in PBS with 0.4% Triton-X 100 (PBST) for 30 min. Primary antibodies were incubated in 1% milk in PBST overnight at 4°C. Secondary antibodies (goat anti-mouse Alexa 488 and donkey anti-rabbit Alexa 546; Invitrogen A-21121 and A10040, respectively) were incubated in 1% milk in PBST for 1 hr. For manual curation, images from at least two independent experiments were analyzed blindly and independently by two investigators, and classified into four non-mutually exclusive categories (broad, narrow, focal and at the heterochromatin boundary) based on the predominant localization patterns within a population of cells.

### Materials and data availability

RNAi screen data are available at the *Drosophila* RNAi Screening center (http://www.flyrnai.org/cgi-bin/DRSC_screen_csv.pl?project_id=151) and the PubChem BioAssay Database, AID= 1159615 (https://pubchem.ncbi.nlm.nih.gov/assay/assay.cgi?aid=1159615). The mass spectrometry proteomics data have been deposited to the ProteomeXchange Consortium via the PRIDE (*Vizcaíno et al., 2016*) partner repository with the dataset identifier PXD003780 and 10.6019/PXD003780.

## Acknowledgements

We thank the *Drosophila* RNAi Screening Center and the Bloomington Stock Center for fly stocks and reagents, James Bentley Brown for helpful discussion regarding the RNAi screen data analysis and general advice on statistical methods, the University of California at Berkeley (UCB) Department of Statistics consulting group for helpful discussions regarding data analysis of the RNAi screen, Cameron D. Kennedy for help on bioinformatics analyses, Ben Bowen and Chris Petzold for their help interpreting the HP1a IP-MS/MS datasets, the High Throughput Screening Facility at UCB for advice and use of equipment regarding high-throughput assays (especially Trish Birk and Mary West), Sue Celniker for the generous gift of clones, Charles Yu and Kenneth Wan for advice on high-throughput cloning, Aniek Janssen, Wilbur Kyle Mills and Grace Yuh Chwen Lee for helpful comments on the manuscript, Mitzi Kuroda for the ADD1-BioTAP plasmid, and Axel Imhof and Andreas Thomae for helpful discussions regarding HP1a complexes.

## Additional information

### Funding

| Funder | Grant reference number | Author |
| --- | --- | --- |
| Lawrence Berkeley National Laboratory | LB11015 | Joel M Swenson Sylvain V Costes Gary H Karpen |

| National Institutes of Health | NRSA Trainee appointment, T32 GM 007232 | Joel M Swenson |
|---|---|---|
| National Institutes of Health | Ruth Kirchstein NIH Postdoctoral Fellowship, 1F32GM086111 | Serafin U Colmenares |
| National Science Foundation | Graduate Research Fellowship, DGE 1106400 | Amy R Strom |
| National Institutes of Health | RO1, GM086613 | Gary H Karpen |
| Reshetko Family Scholarship in the College of Letters & Science | | Joel M Swenson |

The funders had no role in study design, data collection and interpretation, or the decision to submit the work for publication.

### Author contributions

JMS, Conceived the experiments, Interpreted results, Performed purifications, the RNAi screen, Imaging for the RNAi screen and associated data analyses, Subcloned constructs for and performed the low-resolution imaging screen, Performed fly crosses, Acquired data for the PEV assay and performed the associated image analysis and interpretation, Cloned/subcloned constructs and performed high-resolution and time-lapse imaging, Wrote the manuscript; SUC, Conceived the experiments and interpreted results, Performed imaging for the RNAi screen, Cloned/subcloned constructs, Performed fly crosses and acquired data for the PEV assay and cloned/subcloned constructs and performed high-resolution and time-lapse imaging, Edited the manuscript; ARS, Subcloned constructs for and performed the low-resolution imaging screen, Edited the manuscript; SVC, Performed image analysis and the unsupervised clustering for the RNAi screen, Edited the manuscript; GHK, Conceived the experiments, Interpreted results, Wrote the manuscript

### Author ORCIDs

Serafin U Colmenares, http://orcid.org/0000-0002-4094-4220
Gary H Karpen, http://orcid.org/0000-0003-1534-0385

## Additional files

### Supplementary files

• Supplementary file 1. GO enrichment analysis of HPips. 135 HPips were analyzed using DAVID for enrichment of GO terms. The GO enrichment category is shown along with the p-value and the HPips that belong to that GO enrichment category. Some redundant GO categories were removed for ease of viewing by selecting a representative GO category for each Functional Annotation Cluster reported by DAVID. Blue GO enrichment categories indicate that the categories belonged to the same Functional Annotation Cluster. GO categories representing ribosomal subunits and structural constituents of cytoskeleton were enriched, but are not displayed since they are considered common contaminants.

• Supplementary file 2. Description of imaging features extracted from images. Using custom Matlab scripts, the number of nuclei per well were counted and 32 imaging features were measured per nucleus and averaged per well. These averaged imaging features combined with the number of nuclei yields an 'imaging signature' for each well.

• Supplementary file 3. 564 genes putatively down-regulate HP1a. Gene depletions that led to an increase in HP1a mean and max intensity are listed (HP1a negative regulators).

• Supplementary file 4. GO enrichment analysis on all RNAi hits. All hits were analyzed using DAVID for enrichment of GO terms. The GO enrichment category is shown along with the p-value and the gene that belongs to that GO enrichment category. Redundant GO categories were removed by selecting a representative GO category for each Functional Annotation Cluster as reported by

DAVID. Red indicates a GO category that was not significantly enriched but was deemed to be of interest.

• Supplementary file 5. A summary of the data presented in this manuscript. Unless otherwise noted the data presented here was generated as part of this study. Not all proteins that colocalized with HP1a as part of the 'low-resolution' imaging screen were tested for effects on PEV or imaged at higher resolution. If these proteins were previously studied for PEV effects or localization, references are provided. See the main text for details.

• Supplementary file 6. A list of primers used in this manuscript. All sequences are in 5' to 3' orientation; genic sequences are shown in upper case; flanking sequences containing restriction sites are shown in lower case.

## Major datasets

The following datasets were generated:

| Author(s) | Year | Dataset title | Dataset URL | Database, license, and accessibility information |
|---|---|---|---|---|
| Swenson JM, Colmenares SU, Costes SV, Karpen GH | 2016 | Genome-wide RNAi Screen to Identify Regulators of Heterochromatin Recruitment or Maintenance | http://www.flyrnai.org/cgi-bin/DRSC_screen_csv.pl?project_id=151 | Publicly available at the Flyrnai website (Project ID: 151) |
| Swenson JM, Colmenares SU, Costes SV, Karpen GH | 2016 | Genome-wide RNAi Screen to Identify Regulators of Heterochromatin Recruitment or Maintenance | https://pubchem.ncbi.nlm.nih.gov/assay/assay.cgi?aid=1159615 | Publicly available at the the NCBI PubChem website (AID: 1159615) |
| Swenson JM, Colmenares SU, Karpen GH | 2016 | HP1a IP-MS | http://proteomecentral.proteomexchange.org/cgi/GetDataset?ID=PXD003780 | Publicly available at ProteomeXchange (accession no: PXD003780) |

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

## Appendix

## Identification of HP1a interaction partners

To gain a better understanding of the composition of heterochromatin and to further elucidate the role of HP1a and HPips in the DNA damage response in heterochromatin we performed two sets of HP1a purifications at various time points with respect to DNA double-strand break (DSB) induction. HP1a was purified before IR, at 10 min post-IR (HP1a is expanded and γH2Av foci are in heterochromatin), and at 60 min post-IR (Rad51 is loaded onto DSBs at the heterochromatin periphery, HP1a is starting to contract) (*Chiolo et al., 2011*) using S2 cells stably expressing FS-HP1a. To preserve as many HP1a-interacting partners as possible, we performed single-step immunoprecipitations (IP) of FS-HP1a using 3X-FLAG affinity resin (*Table 1—source data 1* and *3*). In addition, tandem affinity purifications were performed using 3X-FLAG and StrepII resins to recover only the most tightly bound HP1a interactors (*Table 1—source data 2*). In both cases purified samples were digested with trypsin and analyzed by liquid chromatography-tandem mass spectrometry (LC-MS/MS) to identify associated proteins.

Based on the bulk of the HPips not being consistently identified at certain time points after IR and the lack of GO pathway enrichment suggesting those HPips are involved in the heterochromatic DNA damage response (DDR), we conclude that bulk HP1a associations do not change significantly in response to DNA damage (*Table 1—source data 1* and *2*), and therefore focused on the rich set of novel HPips identified.

## Data normalization of the RNAi screen

Each of the 120 plates had three mock transfected wells (negative control), four wells transfected with amplicons targeting GFP (negative control), four wells transfected with *thread* amplicons (in the absence of *thread* cells die; control for dying cells/knock-down efficiency) and one or two wells transfected with HP1a amplicons (positive control). Positive and negative controls were used to assay the effectiveness of normalization and statistical analyses.

Various normalization approaches have been suggested to allow comparisons of depletions on one plate to depletions on another (*Birmingham et al., 2009*). In order to optimize the true positive discovery rates and minimize false discovery rates we compared two different normalization techniques: 1. robust z-score (*Chung et al., 2008*) - normalization based on the median well average (wa) of a plate ([wa(sample)- (median(mean of all was))]/median absolute deviation(mean of all wells)) 2. Rank Product (RP) normalization (*Breitling et al., 2004*) - ranks each well in ascending order and multiplies it by the rank of the corresponding well on the replicate plate which yields a RP. The RP was then ranked (RP-rank) to provide a single value for replicates. Additionally, a rank-product statistic (roughly equivalent to a p-value) was estimated for replicates (*Breitling et al., 2004*) and used to identify positive hits.

To determine which normalization technique was optimal we combined all the data for a given normalization technique (robust z-score and RP p-value) for various imaging features (HP1a maximum, HP1a mean and HP1a kurtosis) and plotted the normalized value by rank. We consistently observed that 100% of HP1a controls were detected at a lower rank using the RP p-value compared to the robust z-score. For example, using HP1a mean intensity, all the HP1a controls were detected in the top 2.8% of the screen when normalized by robust z-score, compared to 2.1% by RP p-value (0.7% could represent over 300 unique genes) (*Figure 2— figure supplement 1*). Additionally, we considered the incorporation of replicate consistency and an estimated p-value to be valuable in determining *bona fide* hits. Consequently RP normalization was utilized for all further analyses.

# Identification of hits from the RNAi screen

After data normalization, we utilized three different candidate identification methods to maximize the number of true positive hits. We used the traditional method, specifically ranking a feature of interest, choosing an ad hoc cut-off value and manually inspecting images to confirm phenotypic effects (*Logan and Carpenter, 2010*). We also used more sophisticated approaches (supervised and unsupervised clustering) (*Dürr et al., 2007*; *Jones et al., 2009*; *Loo et al., 2007*), which incorporate different imaging features to detect hits that might otherwise go unobserved. For example, protein depletions that only subtly lower the levels of HP1a but affect all other measured imaging features in the same direction as HP1a depletion would not be detected as a hit using the traditional method. Therefore, to overcome these limitations, supervised machine learning (support vector machine [SVM]) and unsupervised hierarchical clustering approaches were also used.

While HP1a positive controls were identified as hits in all individual HP1a-metric categories, the overlap of novel gene identification was minimal between categories (*Figure 2—source data 1* for a list of all hits). This suggests that there are different classes of genes that affect HP1a levels differently. Unexpectedly, we observed an inverse relationship between dying cells and HP1a mean intensity (*Appendix 1—figure 1*). We manually inspected high scoring hits (e.g. low HP1a mean intensity) and discarded hits where cells were dying (as indicated by distinct DAPI morphology and number of nuclei present).

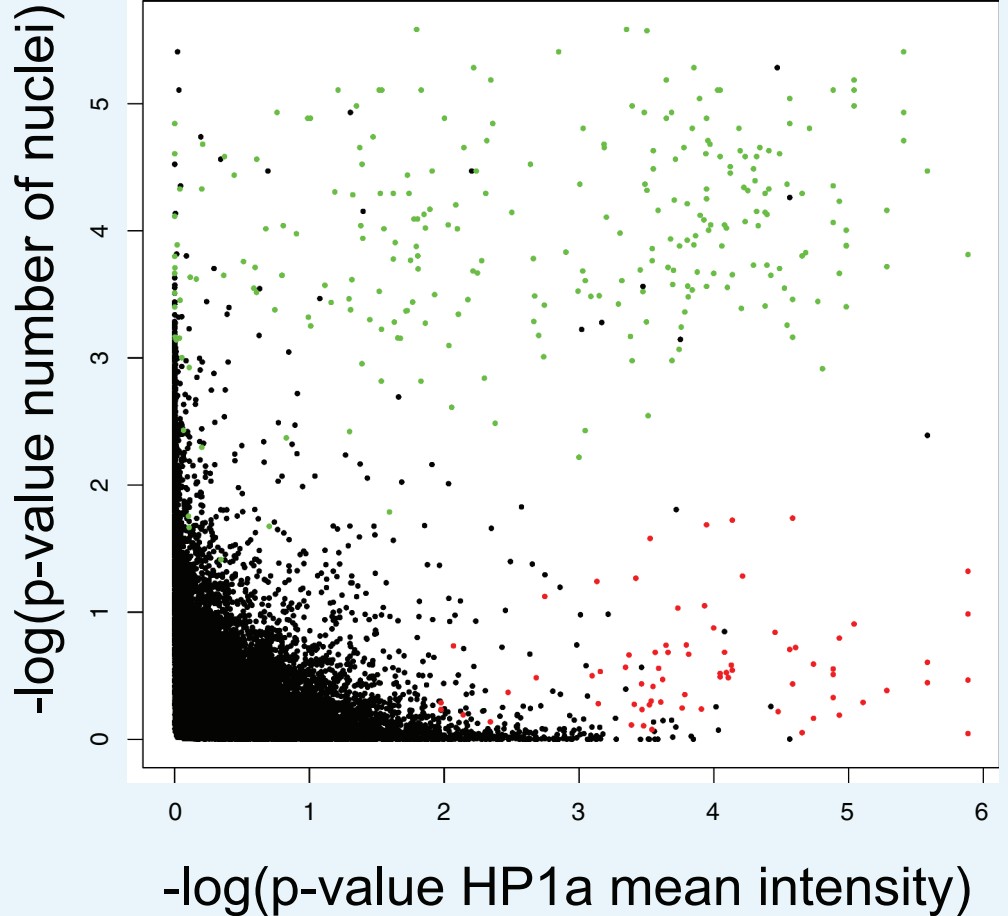

**Appendix 1—figure 1.** HP1a staining is significantly decreased in dying cells. The negative log of the RP p-value for HP1a mean intensity and number of nuclei were plotted against each

other. Green dots indicate thread depletions (dying cells) and red dots indicate HP1a depletions.

Surprisingly, using HP1a-metrics we only identified three genes (*Ino80*, *Parp* and *Adar*) that had previously been shown to affect heterochromatin. The identification of the DNA helicase *Ino80* (part of the Ino80 chromatin remodeling complex) as a hit is consistent with mouse studies where depletion of an Ino80 binding partner (YY1) in mouse spermatogenesis leads to decreased levels of HP1gamma and H3K9me3, and an increase in DNA damage (*Wu et al., 2009*). Studies in *Drosophila* showed that Poly ADP ribose polymerase (Parp) mutations act as enhancers of variegation, cause chromatin to be more nuclease sensitive, and derepress *copia* retrotransposons (*Tulin and Spradling, 2003*). *Adar* is an adenosine deaminase acting on RNA which modifies adenosine to inosine (*Nishikura, 2010*) and has been shown to enhance variegation (*Savva et al., 2013*). The remaining 57 genes identified by this method are promising candidates for regulating HP1a (see *Figure 2—source data 1* for a list of all hits).

In addition to identifying hits through iterative ranking of imaging features, we sought to exploit the high-dimensionality of our data to identify hits through both supervised and unsupervised clustering. Dürr and colleagues (*Dürr et al., 2007*) demonstrated that support vector machines (SVMs) were very effective at classifying positive and negative controls of the supervised classifiers they tested. Here, the SVM was trained using spiked-in HP1a knockdowns for one class and negative controls (mock, GFP, Rho1- binucleate, Thread- cell death) for another class. The classifier was then applied to the entire dataset. The SVM utilized either all imaging features or all imaging features except those associated with γH2Av (denoted "SVM HP1a only features" in *Figure 2—source data 1*).

The small overlap of only four genes between the two different SVMs (*Figure 2—source data 1*) may indicate that either γH2Av-related imaging features provide increased discriminatory power for identifying hits, or are very noisy and obscure the detection of true positives. We favor the former possibility because it has been observed that HP1a is required for the movement of γH2Av foci from heterochromatin (*Chiolo et al., 2011*), and we have also observed that in the absence of HP1a, γH2Av foci are brighter (data not shown). However, hits from each category need to be investigated further to test this hypothesis.

The four overlapping genes (*Figure 2—source data 1*) (CG31673, CG7214, Or47b and bin3) are of special interest for follow-up studies due to their detection by both SVMs. Bin3 contains a S-adenosyl-L-methionine binding domain, may methylate small ncRNAs and interacts with AGO2 to promote post-transcriptional gene silencing (*Singh et al., 2011*). AGO2 has previously been implicated in heterochromatin formation (*Noma et al., 2004*; *Deshpande et al., 2005*). Casein kinase I-alpha was also identified by a SVM. HP1a is heavily phosphorylated during heterochromatin formation and Casein kinase II has been implicated in heterochromatin function (*Zhao and Eissenberg, 1999*). One might speculate that HP1a is also phosphorylated by Casein kinase I and that this PTM is required for HP1a loading. Wapl and modulo have been previously identified as Su(var)s (*Verni et al., 2000*; *Garzino et al., 1992*), and modulo localizes to the chromocenter (*Perrin et al., 1998*) and is required for proper centromere formation (*Chen et al., 2012*) and clustering (*Padeken et al., 2013*). *Crol*, *blanks* and *l(3)neo38* were discovered to regulate heterochromatin-mediated silencing (*Schneiderman et al., 2010*) and both l(3)neo38 and blanks are involved in small RNA pathways (*Zhou, 2008*; *Gerbasi et al., 2011*; *Handler et al., 2013*).

The final approach we took to identify hits was unsupervised hierarchical clustering using four different distance measures (Spearman, Mahalanobis, Euclidean, Pearson). This method identified genes that were pair-wise close to at least five HP1a spike-in knockdowns by more than one distance metric (referred to as 'Clusters with HP1a' in *Figure 2—source data 1*). Surprisingly, Su(var)3-9 did not cluster with HP1a. One possible explanation comes from the observation that Su(var)3-9 knockdown in *Drosophila* tissue culture cells leads to HP1a becoming diffuse across the whole nucleus, without a drastic change in total HP1a levels (data not shown). In contrast, HP1a depletion leads to lower total levels of nuclear HP1a. Thus, we

reasoned that genes required for Su(var)3-9 mediated HP1a deposition might be identified by interrogating genes that displayed pair-wise phenotypes close to Su(var)3-9 depletion. Finally, we constructed neighborhoods based on the Pearson Correlation distance metric using Matlab's dendrogram function (referred to as "HP1a Cluster – Pearson correlation" in *Figure 2—source data 1*) and observed a non-overlapping list of hits compared to pair-wise distance-based measures.

The Su(var)3-9 neighborhood proved adept at identifying genes previously implicated (either directly or indirectly) in heterochromatin assembly or function, including *Hdac3, jumu, MTA1-like, Ssrp, Rm62* and *AGO2*. Hdac3 is a broad specificity histone deacetylase that interacts with the human Su(var)3-9 ortholog (*Vaute et al., 2002*) and in mouse cells deletion of Hdac3 increases H3K9ac, H3K14ac and decreases HP1-beta levels (*Bhaskara et al., 2010*). Jumu is a transcription factor and a haplo-suppressor/triplo-enchancer or haplo-enhancer/triplo-suppressor of PEV, depending on the tissue assayed (*Hofmann et al., 2009*). Overexpression of Jumu leads to a sixfold increase in HP1a transcripts, eviction of Jumu from the polytene chromocenter, and spreading of HP1a (but not H3K9me2) along polytene chromosomes (*Hofmann et al., 2010*). Similar to Su(var)3-9 mutants (*Peng and Karpen, 2009*), misregulation of Jumu (overexpression or loss-of-function mutations) leads to fragmented nucleoli (*Hofmann et al., 2010*). Ssrp (found with all three unsupervised clustering methods) is a subunit of the facilitates chromatin transcription (FACT) complex (*Orphanides et al., 1999*), and in *S. pombe* FACT is implicated in heterochromatin formation (*Lejeune et al., 2007*). AGO2 is part of the RNA-induced transcriptional silencing complex (RITS) complex (*Volpe et al., 2002*; *Irvine et al., 2006*) and has been implicated in the targeting of HP1a to heterochromatin in *S. pombe* and *Drosophila* (*Noma et al., 2004*; *Deshpande et al., 2005*). Additionally, AGO2 has been shown to interact with Tudor-SN in *Drosophila* (*Caudy et al., 2003*). Tudor-SN was found pairwise close to HP1a and is a component of the RNA-induced silencing complex complex in *Drosophila* (*Caudy et al., 2003*). Rm62 is a RNA helicase that interacts with Su(var)3-9, blanks, AGO2 and regulates heterochromatin silencing (*Schneiderman et al., 2010*; *Gerbasi et al., 2011*; *Boeke et al., 2011*; *Ishizuka et al., 2002*; *Csink et al., 1994*). Another hit found pairwise close to Su(var)3-9, ptip, is part of a H3K4 methyltransferase and its depletion in *Drosophila* leads to a decrease in H3K4me3 and an increase in H3K27me3 (*Fang et al., 2009*). Its effects on H3K9 methylation were not addressed.

Additionally, these unsupervised clustering methods identified two components of the nucleosome remodeling and deacetylase (NuRD) complex: MTA1-like (found to cluster with Su(var)3-9) and MBD-like (found to cluster with HP1a). MBD-like can repress a luciferase reporter construct and is localized at the chromocenter in polytene squashes (*Ballestar et al., 2001*). The *Drosophila* NuRD complex also consists of Caf1, Rpd3 (HDAC1) and Mi-2 (*Marhold et al., 2004*). A Caf1 component (p150) is required for HP1a dynamics during replication, replication through heterochromatin, and ultimately cell cycle progression (*Quivy et al., 2008*). Consistent with this, Caf1-depleted wells had significantly fewer cells than other wells, suggesting a cell cycle defect. It is interesting to note that MBD-like and MTA1-like depletions did not lead to fewer cells, suggesting a Caf1-independent role.

Other genes of interest found by unsupervised clustering include Spt20, kismet and Stellate. Spt20 is part of the *Drosophila* SAGA complex (*Weake et al., 2009*) which includes a broad-specificity histone acetyl transferase (HAT), Gcn5, which acetylates H3K9, K14, K18, and K23 (*Kuo and Andrews, 2013*). Spt20 may negatively regulate Gcn5, resulting in an increase in H3K9ac, and consequently a decrease in H3K9me2/3. Kismet is a chromatin remodeler with two chromodomains and an ATPase domain (*Daubresse et al., 1999*), and mutations suppress variegation (*Schneiderman et al., 2009*). Stellate is a Casein kinase II (CK2) subunit (*Bozzetti et al., 1995*) and CK2 has been implicated in heterochromatin formation and may directly phosphorylate HP1a (*Kellum et al., 1995*; *Zhao and Eissenberg, 1999*; *Eissenberg et al., 1994*).

