## [Decision Letter]

[Editors’ note: this article was originally rejected after discussions between the reviewers, but the authors were invited to resubmit after an appeal against the decision.]

Thank you for submitting your manuscript to *eLife* for consideration. In addition to our Board of Reviewing Editor member, we assigned the manuscript to three other experts in the field. After much consideration and discussion among the reviewers, it is our consensus that this manuscript is not appropriate as a research article for *eLife*, and therefore, we are not able to consider it any further. However, the reviewers suggested that the manuscript could be considered as a resource for *eLife*, but will require several changes and further analysis (as outlined in the reviewers' comments) before that study can be reconsidered as a resource. As you can see from the reviewers' comments below, these changes are extensive both textually and experimentally. Also, please consider if you are to submit the new manuscript as a resource, it will go under a full review by the journal.

Reviewer #1:

The molecular biology of heterochromatin has made considerable progress in the past 30 years, owing in large part to the molecular genetic dissection of heterochromatin in fission yeast and *Drosophila*. It has been known for decades that the DNA sequence composition of *Drosophila* heterochromatin is heterogeneous, and so it should be unsurprising that the protein composition of heterochromatin is similarly heterogeneous. The work in this paper extends our appreciation of this heterogeneity.

This manuscript greatly expands the number of proteins that appear to associate directly or indirectly with *Drosophila* HP1a, one of the first heterochromatin-associated proteins to be molecularly and genetically characterized. It identifies a subset of these as genetic modifiers of heterochromatic position-effect silencing by Y chromosome heterochromatin.

Overall, the manuscript is well-written, the studies are well-designed, the data are novel and worth publishing and the conclusions are generally supported by the data. I found the Discussion somewhat overlong and unnecessarily speculative.

Reviewer #2:

The manuscript addresses the fundamental issue of heterochromatin composition, structure, and dynamics. The authors apply a very broad range of modern scientific technologies combined with methodical data analyses to identify new components of heterochromatin. They found a number of genes whose products contribute to HP1a complexes and/or whose functions affect the distribution patterns of HP1a in the cell nucleus, identifying some previously known and many novel contributors. The results were obtained by conducting two different types of high throughput screens (biochemical – HP1a binding – and genetic – perturbation of distribution patterns in mutants) in an unbiased manner. The volume of the data presented in this manuscript is very impressive. While the results per se do not lead to a paradigm shift (the idea of subdomains within heterochromatin has been previously described), they potentially can trigger a whole series of new investigative projects, as biomedical scientists spot some of their favorite proteins on the list. Introduction and maintenance of gene silencing by heterochromatin formation is increasingly being recognized as a key mechanism in differentiation and maintenance of health (i.e. avoiding cancer), making this a very active area of research.

The information in this manuscript could have been parsed into more than one paper. The authors' choice was to incorporate all of the materials into one publication, and this certainly has advantages in appreciating the scope of the diversity observed in HP1a interacting proteins. However, we recommend reducing the volume of the manuscript to keep the reader focused. In particular, both the Introduction and later text have a wide range of information that does not seem to be related to the results presented here. This content could be significantly condensed and the remaining text used for a review (published elsewhere) or transferred to the supplementary materials. On the other hand, the choice of figures to include in the main text seemed appropriate, and no changes are recommended in this regard.

Other considerations:

The authors should describe some of the particular vocabulary they use (e.g. "HP1a morphology") when first introduced, and spell out abbreviations when first introduced into the text.

The authors need to provide a list and a source of all stocks used in the genetic experiments (PEV analysis).

Reviewer #3:

The manuscript "The composition and organization of *Drosophila* heterochromatin are heterogeneous and dynamic" describes the results of two screens for heterochromatic proteins in *Drosophila* cells. The first uses co-immunoprecipitation with the heterochromatic protein HP1a followed by mass-spectrometry to identify 118 new putative HP1-interacting proteins. A second cytological screen identifies 374 genes that when knocked-down by RNAi reduce HP1a localization within the nucleus. There is little overlap between the results of these two screens. It is not clear how some factors are interpreted as noise while others are considered candidate factors, which is a critical aspect of such screens.

This manuscript lacks a conceptual framework for understanding the localizations and genetic effects of the candidate factors. Suggestions are made in Figure 8, but without experimental evidence this figure should be deleted.

The presentation of the results of these screens is disorganized, especially as most assays are not comprehensively applied to all the candidate factors (presumably decided as reagents were available?). A table summarizing the assays that were applied to candidates and their results would help the organization. Organizing the figures would also help, as figures jump through a long list of factors. Figure 3 presents two heterochromatic proteins; different ones are presented in Figure 5 when types are proposed. In some cases, the summary Figure 7 differs from the primary data presented in the supplement (XNP is characterized as 'narrow' in Figure 7, but as 'narrow' and 'focal' in the supplement; the localization of ADD1 to mitotic chromosomes is very weak in the supplemental images, and it is not explained how is it assessed as localized to mitotic heterochromatin).

---

## [Author Response]

[Editors’ note: the author responses to the first round of peer review follow.]

*Reviewer #1:*

The molecular biology of heterochromatin has made considerable progress in the past 30 years, owing in large part to the molecular genetic dissection of heterochromatin in fission yeast and Drosophila. It has been known for decades that the DNA sequence composition of Drosophila heterochromatin is heterogeneous, and so it should be unsurprising that the protein composition of heterochromatin is similarly heterogeneous. The work in this paper extends our appreciation of this heterogeneity.

We agree that DNA sequence heterogeneity in heterochromatin is well-established. However, the majority of heterochromatin proteins identified and studied here, and elsewhere, do not appear to bind specific sequences. Most importantly, prior to our study, the prevalence of heterogeneous protein localization patterns in heterochromatin was not clear, making our demonstration that many form sub-domains novel.

*This manuscript greatly expands the number of proteins that appear to associate directly or indirectly with Drosophila HP1a, one of the first heterochromatin-associated proteins to be molecularly and genetically characterized. It identifies a subset of these as genetic modifiers of heterochromatic position-effect silencing by Y chromosome heterochromatin.*

Overall, the manuscript is well-written, the studies are well-designed, the data are novel and worth publishing and the conclusions are generally supported by the data. I found the Discussion somewhat overlong and unnecessarily speculative.

We thank the reviewer for the overall positive comments, and have reduced the amount of speculation in the Discussion and the overall length of the Discussion by ~1/2.

*Reviewer #2:*

The manuscript addresses the fundamental issue of heterochromatin composition, structure, and dynamics. The authors apply a very broad range of modern scientific technologies combined with methodical data analyses to identify new components of heterochromatin. They found a number of genes whose products contribute to HP1a complexes and/or whose functions affect the distribution patterns of HP1a in the cell nucleus, identifying some previously known and many novel contributors. The results were obtained by conducting two different types of high throughput screens (biochemical – HP1a binding – and genetic – perturbation of distribution patterns in mutants) in an unbiased manner. The volume of the data presented in this manuscript is very impressive. While the results per se do not lead to a paradigm shift (the idea of subdomains within heterochromatin has been previously described), they potentially can trigger a whole series of new investigative projects, as biomedical scientists spot some of their favorite proteins on the list. Introduction and maintenance of gene silencing by heterochromatin formation is increasingly being recognized as a key mechanism in differentiation and maintenance of health (i.e. avoiding cancer), making this a very active area of research.

We are glad that Reviewer #2 understands the broad impact of our work. We agree that a few cases of subdomains (~6) within heterochromatin were previously described. However, the generality of subdomain architecture for numerous, diverse heterochromatin proteins was not established. More importantly, the fact that this restricted localization is unexpected, and the implications of restricted localization of proteins that directly bind a common, dispersed component (HP1a), has not been discussed in the literature, including the publications that reported on the localization of the 6 proteins. In addition, half of the previous studies were done using polytene chromosomes with under-replicated heterochromatic DNA, which restricts the resolution needed to effectively localize heterochromatin proteins. Our work (using non-polytenized, dividing cells) extends the number of subdomain forming proteins from 6 to 19. Additionally, most previously identified subdomain forming proteins could be classified as “focal”. We identified two other classes “narrow” and “at the heterochromatin boundary”. Finally, we demonstrated using time-lapse imaging that the localization patterns of 5 (out of 7 tested) of the subdomain forming proteins are dynamic throughout the cell cycle. Overall, our findings greatly extend knowledge of the prevalence and dynamic nature of subdomain organization in heterochromatin, emphasizes the importance of future studies to determine how this unexpected architecture is established and maintained, and impacts heterochromatin functions.

The information in this manuscript could have been parsed into more than one paper. The authors' choice was to incorporate all of the materials into one publication, and this certainly has advantages in appreciating the scope of the diversity observed in HP1a interacting proteins. However, we recommend reducing the volume of the manuscript to keep the reader focused. In particular, both the Introduction and later text have a wide range of information that does not seem to be related to the results presented here. This content could be significantly condensed and the remaining text used for a review (published elsewhere) or transferred to the supplementary materials. On the other hand, the choice of figures to include in the main text seemed appropriate, and no changes are recommended in this regard.

We thank Reviewer #2 for the overall positive comments regarding the science and appropriateness of the figures. We agree that the manuscript would benefit from limiting the scope of the Introduction to relevant points, and from “streamlining” the text throughout. We decreased the Introduction by ~1/4 and decreased the Discussion by ~1/2.

*Other considerations:*

The authors should describe some of the particular vocabulary they use (e.g. "HP1a morphology") when first introduced, and spell out abbreviations when first introduced into the text.

We thank Reviewer #2 for bringing these important oversights to our attention. We apologize for not adequately defining “HP1a morphology” and have changed the wording throughout the manuscript to either “phenocopies HP1a depletion” or “mimicked HP1a depletion” or “heterochromatin architecture (e.g. HP1a levels or organization)”. Additionally, we verified that all abbreviations are now defined.

The authors need to provide a list and a source of all stocks used in the genetic experiments (PEV analysis).

The stocks were listed in Figure 4—figure supplement 1 (now called [Supplementary-material SD6-data]), and the sources (i.e. stock centers) were listed in the acknowledgements. To clarify further, we have added the following text to the Materials and methods section: “see Figure 4 and [Supplementary-material SD6-data] for list of fly stocks). Mutant and RNAi fly stocks were all obtained from the Bloomington Stock center, except for Ago2[51B] which was a kind gift from F.B. Gao (Xu et al., 2004).”

*Reviewer #3:*

The manuscript "The composition and organization of Drosophila heterochromatin are heterogeneous and dynamic" describes the results of two screens for heterochromatic proteins in Drosophila cells. The first uses co-immunoprecipitation with the heterochromatic protein HP1a followed by mass-spectrometry to identify 118 new putative HP1-interacting proteins. A second cytological screen identifies 374 genes that when knocked-down by RNAi reduce HP1a localization within the nucleus. There is little overlap between the results of these two screens. It is not clear how some factors are interpreted as noise while others are considered candidate factors, which is a critical aspect of such screens.

We agree with the reviewer that the thoughtful determination of candidate factors is an important part of all screens. However, candidate selection for the HP1a IP-MS and the RNAi screen were discussed in depth in the Materials and methods section. We did not definitively interpret any factors as “noise” based on the IP-MS or RNAi screen alone. However, to reduce the scale of the follow-up experiments we did remove common contaminants (e.g. ribosomal proteins) (subsection “A subset of IP-MS and RNAi hits colocalize with HP1a”, first paragraph) while cautioning that these are potentially biologically relevant. We now include a reference justifying the removal of common contaminants from IP-MS experiments (in the aforementioned paragraph). Additionally, to further reduce the scale of the follow-up experiments we selected a subset of the RNAi hits based on GO terminology enrichment and availability of reagents (in the aforementioned paragraph).

We would appreciate some additional guidance from the editor or reviewer to ensure that this issue is adequately addressed.

This manuscript lacks a conceptual framework for understanding the localizations and genetic effects of the candidate factors. Suggestions are made in Figure 8, but without experimental evidence this figure should be deleted.

We are not sure what kind of conceptual framework the reviewer is referring to, but are very open to incorporating appropriate changes with further clarifying input. Figure 8 represents our attempt to provide ideas for how subdomains could form despite the uniform distribution of the HP1a binding partner, and was not meant to be a figure that summarizes our findings. Although we believe that readers would appreciate a clear representation of the hypotheses, we are willing to remove Figure 8 if that is a consensus. However, note that Reviewer #1 had no comment in this issue and that Reviewer #2 explicitly stated “the choice of figures to include in the main text seemed appropriate, and no changes are recommended in this regard.”

*The presentation of the results of these screens is disorganized, especially as most assays are not comprehensively applied to all the candidate factors (presumably decided as reagents were available?).*

Throughout the manuscript we attempted to be as clear as possible regarding the rationale for which candidate factors (out of ~500 genes) were subjected to follow-up analysis. Here is an outline of how we narrowed down candidate factors: Figure 1 is a workflow figure outlining our overall strategy. In the first paragraph of the subsection “A subset of IP-MS and RNAi hits colocalize with HP1a” we describe how candidates from the IP-MS and RNAi screen were selected for further analysis (the low-resolution imaging screen) and state that reagent availability was a factor. In the subsection “Many HPips and HPprs affect heterochromatin-mediated silencing” it is stated that we assayed for modifiers of PEV based on the availability of reagents (mutations and RNAi lines). In the subsection “Localization of IP-MS and RNAi hits reveals complex patterns within heterochromatin” we clarify that we chose “a subset (19) of the top HP1a colocalization hits (30)” to analyze at higher-resolution. In the first paragraph of the subsection “Live imaging reveals that subdomain protein localization patterns are dynamic” we state that we performed time-lapse imaging on 7 proteins that displayed more than one localization pattern in our previous studies. We request more guidance from the editor or reviewer if further improvements are required.

*A table summarizing the assays that were applied to candidates and their results would help the organization.*

We thank Reviewer #3 for this idea and have added a summary table as [Supplementary-material SD11-data].

*Organizing the figures would also help, as figures jump through a long list of factors. Figure 3 presents two heterochromatic proteins; different ones are presented in Figure 5 when types are proposed.*

We thank Reviewer #3 for suggesting revisions to Figure 3. We were attempting to show a diverse set of examples, and have now modified Figure 3 to include the same heterochromatin proteins as shown in Figure 5.

*In some cases, the summary Figure 7 differs from the primary data presented in the supplement (XNP is characterized as 'narrow' in Figure 7, but as 'narrow' and 'focal' in the supplement;*

We thank Reviewer #3 for pointing this out. XNP is actually characterized as ‘narrow’ and ‘focal’ in Figure 7 as well, but we agree that it should be clearer. We placed lines around ‘focal’ subdomains in Figure 7 to ensure that ‘focal’ subdomains can be seen when combined with ‘narrow’ subdomains.

*the localization of ADD1 to mitotic chromosomes is very weak in the supplemental images, and it is not explained how is it assessed as localized to mitotic heterochromatin).*

We agree with Reviewer #3 that ADD1’s localization to mitotic chromosomes is very weak (subsection “Live imaging reveals that subdomain protein localization patterns are dynamic”, third paragraph; Figure 7—figure supplement 2 legend, Video 3 legend) and are willing to remove the statement saying that it remains attached to chromatin if they feel the evidence is not convincing, but believe it is worth including (with appropriate qualifiers) since it was consistently observed across multiple experiments, and is unusual for HP1a interacting proteins, which in all other cases to date are removed with HP1a during mitosis. A minor, but important clarifying point: we did not state that ADD1 is localized to mitotic *heterochromatin*, just to mitotic chromosomes.